# AHE-FNUQ: An Advanced Hierarchical Ensemble Framework with Neural Network Fusion and Uncertainty Quantification for Outlier Detection in Agri-IoT

**DOI:** 10.3390/s25226841

**Published:** 2025-11-08

**Authors:** Ahmed Amamou, Mimoun Lamrini, Bilal Ben Mahria, Younes Balboul, Said Hraoui, Omar Hegazy, Abdellah Touhafi

**Affiliations:** 1IASSE Laboratory, Computer Science Department, National School of Applied Sciences, Sidi Mohamed Ben Abdellah University, Fez 30050, Morocco; younes.balboul@usmba.ac.ma (Y.B.); said.hraoui@usmba.ac.ma (S.H.); 2MOBI-EPOWERS Research Group, ETEC Department, Vrije Universiteit Brussel (VUB), 1050 Brussels, Belgium; omar.hegazy@vub.be; 3Flanders Make, 3001 Heverlee, Belgium; 4Computer Science Department, Faculty of Science and Technology, Sidi Mohamed Ben Abdellah University, Fez 30000, Morocco; bilal.benmahria@usmba.ac.ma; 5Department of Engineering Sciences and Technology (INDI), Vrije Universiteit Brussel (VUB), 1050 Brussels, Belgium; abdellah.touhafi@vub.be

**Keywords:** IoT, ECOD, COPOD, HBOS, OC-SVM, Isolation Forest, KNN, anomaly detection, agricultural IoT

## Abstract

Agricultural Internet of Things (Agri-IoT) systems need strong anomaly detection to monitor crops effectively. However, current approaches lack accuracy and efficiency. To mitigate this problem, we proposed an advanced hierarchical ensemble framework with neural network fusion and uncertainty quantification (AHE-FNUQ). This framework combines six detection algorithms: Isolation Forest, ECOD (empirical cumulative distribution-based outlier detection), COPOD (copula-based outlier detection), HBOS (histogram-based outlier score), OC-SVM (one-class support vector machine), and KNN (k-nearest neighbors). It uses a three-level decision process: (1) selecting models with good performance (ROC AUC > 0.75), (2) applying recall-weighted ensemble fusion, and (3) using a fusion neural network (FusionNN) to improve uncertain predictions in the confidence range [0.75,0.9]. The framework was tested on three agricultural datasets with contamination levels between 10% and 50%. The result showed strong performance: ROC AUC between 0.93 and 0.99, PR AUC between 0.90 and 0.98, and F1-scores between 0.85 and 0.90. Moreover, we have conducted a statistical test (Friedman test, $\chi^{2}=63.02$, p<0.0001) and confirmed that AHE-FNUQ is significantly better than common methods such as COPOD, ECOD, HBOS, Isolation Forest, and KNN.

## 1. Introduction

Anomaly detection is a crucial problem in machine learning (ML) that has gained attention in many fields, such as cybersecurity [1,2,3], healthcare [4,5,6], industrial monitoring [7,8,9], and financial fraud detection [10,11,12]. The main goal is to find data patterns that are very different from normal behavior. As data grows in size and complexity, single models often fail to capture these patterns, especially when anomalies appear across many dimensions and over time. This is a crucial challenge in agriculture, where anomaly detection supports crop health monitoring, yield prediction, and resource management [13].

Detecting anomalies using supervised learning techniques is challenging as they appear rarely and result in class imbalance problems [14,15]. Since normal behaviors change with time and context, adaptive methods are needed to keep the models up-to-date. The large number of features in modern datasets also leads to the “curse of dimensionality,” where distance-based methods become unreliable [16,17]. Regardless of the promising results shown using the ensemble methods for anomaly detection, they still face several challenges [18,19,20]. Many rely on simple rules, such as majority voting or weighted averaging, which cannot capture complex interactions between models. They also use fixed thresholds that are not well-suited to each detector, reducing accuracy. Most importantly, they do not handle uncertainty well when models disagree, although these are the cases where the ensemble methods should provide the highest benefit [21,22,23,24]. For these reasons, there is a need for a framework that integrates the strengths of ensemble learning methods [25,26], neural networks, and uncertainty quantification [27,28]. Such a system should be accurate, reliable, and efficient for real-world applications.

Based on recent work on neural network fusion [29,30,31,32] and uncertainty quantification [33,34,35], neural networks can combine outputs of detectors in more effective ways, while uncertainty quantification methods can estimate the confidence of predictions. This is especially important in real applications, where false positives or false negatives can be costly. Likewise, hierarchical approaches [36,37,38] have shown their effectiveness in different levels of analysis depending on the complexity of the data.

Existing anomaly detection methods [18,19,20,27,38,39] fail to address the unique challenges of agricultural sensor data, including seasonal variations, irregular sampling patterns, and crop-specific normal behaviors. They were designed for industrial systems and unable to distinguish between natural agricultural variations and actual anomalies.

As a result, this paper focuses on developing a framework that accurately detects true anomalies while accounting for the natural variability inherent in agricultural environments. Hence, we introduce AHE-FNUQ, a novel framework designed to effectively detect and identify anomalies in agricultural data generated by sensor networks. This framework integrates data preprocessing, model selection, adaptive thresholds, and a neural network to handle prediction uncertainty. It uses a hierarchical decision process that balances accuracy, interpretability, and efficiency. To evaluate our approach, we conducted experiments on three independent agricultural datasets using cross-validation. The results show that AHE-FNUQ performs more accurately and consistently than five commonly used baseline methods.

The remainder of this paper is organized as follows. In Section 2, we present the related work that describes the most popular studies on IoT and anomaly detection. Section 3 details the proposed approach. In Section 4, we compare existing approaches with our proposed AHE–FNUQ framework, highlighting methodological differences and advantages. Section 5 describes the experimental setup. Section 6 is dedicated to the presentation and discussion of the experimental results. Finally, we conclude the paper in Section 7 and suggest potential directions for future work.

## 2. Related Work

Anomaly detection in Agri-IoT systems has become a critical research area, driven by the growing use of precision agriculture technologies and the need for reliable monitoring of crop conditions, environmental parameters, and equipment performance. In this section, we provide a structured review of the existing literature across three main categories:**General IoT anomaly detection**, which focuses on generic detection methods applicable to a variety of IoT contexts.**Industrial IoT (IIoT)**, where anomaly detection is essential for ensuring safety and operational efficiency in manufacturing and infrastructure systems.**Agricultural IoT (Agri-IoT)**, a rapidly growing field in which environmental complexity, biological variability, and data sparsity create unique detection challenges.

### 2.1. Anomaly Detection in IoT System

Foundational research in IoT anomaly detection was conducted in previous studies [40], which focused specifically on IoT time-series anomaly detection. The study noted that “the majority of current anomaly detection methods are highly specific to the individual use case, requiring expert knowledge.” This survey highlighted that the nature of IoT data presents significant challenges for applying traditional anomaly detection techniques, particularly due to the relative novelty of IoT applications and the limited availability of established methods.

Building on this foundational research, subsequent studies [27,38] explored outlier detection in sensor data and indoor localization using ML, highlighting the increasing importance of ML approaches in IoT anomaly detection.

Furthermore, a comprehensive review [36] presented ML and deep learning (DL) techniques for IoT anomaly detection, providing the foundation for advanced approaches. Specialized research [28] focused on RNN-based anomaly detection models for IoT networks, illustrating the evolution toward deep learning methods.

More recently, research [41] provided a detailed analysis of current trends and challenges in IoT data anomaly detection. Key issues include high dimensionality, scalability limitations in real-time processing, imbalanced data distributions, and difficulties in interpretability. Current methods also face significant computational complexity with large datasets and require careful parameter tuning. Limited work exists on intelligent ensemble approaches specifically designed for agricultural IoT contexts.

Additionally, a detailed examination [42] of ML and DL techniques for IoT network anomaly detection identified limitations such as algorithm inaccuracy, limited consideration of adversarial attacks, difficulty handling IoT data variability, and insufficient evaluation frameworks. The study emphasizes the need for diverse datasets, real-time testing, and scalable architectures.

Finally, a review [39] of modern anomaly detection for distributed IoT systems identified limitations including reliance on large labeled datasets and limited adaptability. Robust real-time methods that reduce dependence on extensive training data through active learning are needed.

Overall, Table 1 provides a comparative summary of key studies across different IoT domains.

### 2.2. Industrial IoT Anomaly Detection

IIoT systems have attracted significant attention due to their critical role in manufacturing and safety applications. Research in this domain has evolved from basic monitoring techniques to sophisticated ML approaches.

Initial industrial monitoring systems [8] focused on process monitoring using IoT. Subsequent studies [7,9] extended this work to fault detection and the monitoring of industrial electrical equipment.

Recent comprehensive studies [43] provide systematic analyses, revealing that existing mapping studies primarily focus on network and cybersecurity issues, with limited attention to specific industrial sectors and ML challenges.

Research examining anomaly classification [44] highlights significant data quality challenges from incomplete, unstructured, redundant, and noisy data. Implementing anomaly detection in industrial contexts remains challenging, with automatic classification still an open research problem.

Table 2 summarizes the domain-specific challenges and requirements across general, industrial, and Agri-IoT systems.

### 2.3. Agri-IoT Systems and Anomaly Detection

IoT applications in agriculture present unique challenges compared with other domains. Research has evolved from basic implementations to sophisticated DL approaches.

Early studies [45] highlighted the need to manage “spatial data, highly varying environments, task diversity, and mobile devices.” Limitations included affordability, power consumption, network latency, and resilience against extreme weather.

Specialized research [13] focused on anomaly detection in streaming agricultural data, emphasizing the challenges of dynamic environmental conditions.

Subsequent studies [47,48,49,50] highlighted challenges in resource management, Big Data platform design, data heterogeneity, computational demands, and greenhouse monitoring, emphasizing the need for robust, adaptive systems.

Most recently, [46] analyzed DL-based anomaly detection for precision agriculture, noting “predominant reliance on visual data” and the need for large-scale datasets. Anomalies range from plant diseases in images to environmental fluctuations in time-series data.

Research on transfer learning indicates that “in agriculture, data is often sparse due to vast farm areas and cost constraints,” presenting challenges for models that require dense, continuous data streams.

### 2.4. Research Gaps and Limitations

Critical gaps in Agri-IoT anomaly detection include the following:

**Dataset Limitations:** A scarcity of large-scale, diverse datasets hinders progress. Multi-year collections are needed to capture seasonal variations, slowing research and complicating validation.

**Algorithm Specificity:** Existing methods struggle with dynamic agricultural environments. Adaptability is crucial due to biological complexity and environmental variability.

**Real-time Processing Constraints:** Agricultural systems must balance speed with power constraints, especially in remote locations, unlike industrial contexts requiring millisecond responses.

**Transfer Learning Challenges:** Agricultural anomalies differ fundamentally from industrial or urban anomalies, making conventional transfer learning less effective.

### 2.5. Positioning of Proposed Method

Considering these limitations, there is a clear research gap for ensemble-based anomaly detection methods tailored to agricultural IoT. The proposed adaptive hybrid ensemble with fusion neural uncertainty quantification (AHE-FNUQ) addresses these challenges:

**Sparse Data Handling.** The ensemble leverages multiple base detectors, improving accuracy with limited data.

**Seasonal Adaptation.** The hybrid ensemble adapts to seasonal variations, outperforming single-algorithm methods.

**Uncertainty Quantification.** Fuzzy neural uncertainty provides confidence measures, improving interpretability and aiding decision-making.

**Cost-effectiveness.** The ensemble achieves robust performance without the computational cost of DL models, suitable for resource-limited environments.

Table 3 summarizes the research gaps and corresponding considerations addressed by the proposed method.

## 3. Proposed Approach: Advanced AHE-FNUQ for Agricultural Anomaly Detection

### 3.1. Proposed Approach: Advanced AHE-FNUQ Framework

This section gives an overview of the advanced AHE-FNUQ framework, developed to address anomaly detection challenges in Agri-IoT systems, Figure 1.

Agricultural IoT (Agri-IoT) systems operate under highly variable conditions that differ fundamentally from industrial or urban monitoring contexts. They are distributed across large geographic areas, characterized by sparse and noisy sensor coverage, strong seasonal variations, and continuously evolving environmental baselines. In this setting, the cost asymmetry of errors is pronounced: undetected anomalies such as irrigation failures or pest outbreaks may lead to major crop losses, whereas false alarms generally incur minor operational overhead.

Conventional ensemble-based detectors often fail to meet these constraints. They apply uniform aggregation to all predictions regardless of model disagreement, and typically lack mechanisms for quantifying uncertainty. As a result, they cannot indicate when an output should be trusted or re-examined, an essential capability in decision-critical agricultural operations.

The AHE-FNUQ framework addresses these deficiencies through a hierarchical, confidence-aware ensemble architecture that integrates adaptive model selection, recall-weighted score fusion, and uncertainty-driven neural meta-learning. The design enables the system to adjust inference depth according to prediction confidence: simple cases are processed by lightweight aggregation, while ambiguous cases are escalated to a neural meta-learner. This strategy balances accuracy, interpretability, and computational efficiency for deployment on low-power agricultural gateways.

The proposed framework is structured around five interconnected components that collectively enhance anomaly detection accuracy and efficiency. The heterogeneous ensemble strategy integrates multiple detectors to identify diverse anomaly types in agricultural sensor data. The performance-based dynamic selection mechanism selects the most effective models according to objective performance metrics. The recall-weighted fusion combines prediction scores while prioritizing models with higher anomaly detection sensitivity. The three-tier decision system refines uncertain predictions through neural meta-learning to ensure reliability. Finally, the robust preprocessing pipeline standardizes and stabilizes sensor data to maintain consistency under variable environmental conditions. In the following section, each component will be explained in detail.

### 3.2. System Architecture: Step-by-Step Processing Layers

The advanced AHE-FNUQ system processes agricultural sensor data using five sequential layers. Each layer builds on the results of the previous one, creating a structured workflow that enhances anomaly detection performance and reliability, as shown in Figure 2.

#### 3.2.1. Layer 1: Data Input and Robust Preprocessing

This foundational component processes raw sensor readings using mathematical techniques tailored to agricultural environments. Agricultural monitoring data frequently display significant fluctuations resulting from weather variability, equipment calibration changes, and signal interference. These factors necessitate the use of customized data preparation strategies.


**Robust Preprocessing Pipeline**
Normalization is performed using the RobustScaler, which scales data by subtracting the median and dividing by the interquartile range (IQR), defined as the difference between the 75th percentile (Q3) and the 25th percentile (Q1). This approach reduces the influence of extreme values. In contrast, StandardScaler uses the mean and standard deviation, and MinMaxScaler relies on the minimum and maximum values.RobustScaler is selected because agricultural data frequently display substantial variabilities due to seasonal changes, weather patterns, and biological cycles. The objective of anomaly detection is to identify issues such as plant stress, pest infestations, irrigation failures, equipment malfunctions, and atypical environmental conditions, rather than only extreme outliers. StandardScaler may be affected by normal extremes, such as temperature ranges from −5 °C to 45 °C or humidity from 20% to 95%, reducing normalization effectiveness and anomaly detection performance. RobustScaler facilitates the identification of subtle abnormal patterns and provides greater stability in dynamic environments.The mathematical formulation is as follows:(1)$$X_{scaled}=\frac{X-\mathrm{median}(X)}{\mathrm{IQR}(X)}$$
where $\mathrm{IQR}\left(X\right)=Q_{3}\left(X\right)-Q_{1}\left(X\right)$. This method preserves the distinction between normal and abnormal data while ensuring effective normalization with variable agricultural data.Other statistical steps in preprocessing address sensor drift and environmental changes. Missing data is filled using robust techniques, and smoothing algorithms reduce short-term sensor noise while preserving genuine anomaly patterns.
**Enhanced Outlier Generation Strategy**
For model development and evaluation, this layer integrates a dedicated outlier generation mechanism designed to simulate realistic anomaly scenarios in agricultural sensor networks. The goal is to construct labeled datasets that reflect actual failure patterns and environmental disturbances without artificially inflating detection performance.The generation process focuses on producing representative deviations based on agricultural domain characteristics. Two main categories are considered: (i) *subtle anomalies*, representing gradual sensor drifts or mild perturbations, and (ii) *extreme anomalies*, reflecting sudden sensor malfunctions or severe environmental events. Approximately 30% of anomalies affect multiple correlated features simultaneously, reflecting how malfunctions often propagate across dependent sensor measurements in real deployments.Algorithm 1 formalizes this procedure. It randomly selects a subset of sensor samples to simulate events with natural variability. Deviations are applied with magnitudes corresponding to realistic agricultural sensor behavior. No optimization or tuning is applied to favor the model; the purpose is solely to replicate real-world conditions.

**Algorithm 1** Enhanced Agricultural Outlier Generation (Realistic Simulation)
**Require:** Dataset *D* with *n* samples, contamination rate ρ, feature set *F***Ensure:**
 Contaminated dataset $D^{\prime }$ with outlier labels *L*
 num_outliers←⌊n×ρ⌋ outlier_indices← randomly select num_outliers from {1,2,…,n} L← initialize label vector with zeros L[outlier_indices]←1 **for** each feature $f_{i}\in F$ **do**     $\sigma_{i}\leftarrow$ standard deviation of $f_{i}$     subtle_count←⌊num_outliers/2⌋     extreme_count←num_outliers−subtle_count     **for** j=1 to subtle_count **do**         α← random value from U(1.0,1.5)         sign← random choice from {−1,+1}         $D^{\prime }\left[outlier\_indices\left[j\right],f_{i}\right]\leftarrow D\left[outlier\_indices\left[j\right],f_{i}\right]+\alpha \times \sigma_{i}\times sign$     **end for**     **for** j=subtle_count+1 to num_outliers **do**         α← random value from U(2.0,3.0)         sign← random choice from {−1,+1}         $D^{\prime }\left[outlier\_indices\left[j\right],f_{i}\right]\leftarrow D\left[outlier\_indices\left[j\right],f_{i}\right]+\alpha \times \sigma_{i}\times sign$     **end for** **end for** multi_dim_count←⌊0.3×num_outliers⌋ corr_indices← randomly select multi_dim_count from outlier_indices **for** each index k∈corr_indices **do**     $(f_{1},f_{2})\leftarrow$ randomly select a correlated feature pair from *F*     β← random value from U(1.2,1.5)     $D^{\prime }\left[k,f_{2}\right]\leftarrow D^{\prime }\left[k,f_{1}\right]\times \beta$ **end for** **return** $D^{\prime }$, *L*


To illustrate the practical impact of this robust preprocessing pipeline on typical agricultural sensor data, a real-world example is presented in Figure 3.

#### 3.2.2. Layer 2: Multi-Algorithm Detection Ensemble

The detection core integrates six anomaly detection methods, each based on a distinct algorithmic paradigm. The ensemble applies a learning strategy designed to address the heterogeneous nature of agricultural anomaly patterns. No single algorithm achieves optimal performance across all anomaly types or data distributions in Agri-IoT environments.


**Base Detector Selection and Algorithmic Diversity**
The ensemble comprises six detection algorithms, each chosen to address specific characteristics of anomalies in agricultural sensor data:**Isolation Forest:** This tree-based isolation method works well in high-dimensional spaces through random partitioning. It effectively detects global anomalies and handles mixed-type data common in agricultural sensor networks. Anomalies are isolated with fewer splits in decision trees, making IForest computationally efficient for large-scale agricultural monitoring.**ECOD:** A parameter-free statistical method using empirical distribution functions. ECOD performs robustly across different data types and excels at univariate outlier detection. It is computationally efficient, suitable for real-time agricultural monitoring.**COPOD:** This algorithm models complex multivariate data dependencies using the copula theory. It captures intricate relationships and multivariate anomalies, common in interconnected agricultural sensor networks where environmental factors exhibit complex interdependencies.**HBOS:** HBOS uses histogram-based probability density functions for efficient density estimation. It is suitable for large datasets and real-time processing in precision agriculture. HBOS effectively detects density-based anomalies in agricultural time-series data.**KNN:** This proximity-based method identifies anomalies through distance metrics in feature space. KNN is effective for detecting local density deviations and contextual anomalies within sensor clusters, such as gradual sensor drift patterns typical in agricultural environments.**OC-SVM:** This boundary-based method learns a hypersphere around normal data patterns. It is effective at detecting anomalies outside the normal boundaries, particularly for non-linear patterns in complex agricultural sensor data with seasonal variations.

#### 3.2.3. Layer 3: Performance-Based Model Selection

This layer implements a dynamic evaluation mechanism that measures the performance of each detector based on model quality and relevance. The goal is to select the most suitable models for the ensemble, maximizing overall system performance.


**Dynamic Selection Framework**

**Selection Criteria:**
–ROC AUC >0.75: This threshold reflects a model’s strong ability to distinguish between normal and anomalous instances across different decision levels.–Average Precision Score >0.75: This threshold ensures reliable performance on imbalanced datasets, which are common in agricultural anomaly detection.**Computational Optimization:** Only models that meet the established benchmarks contribute to ensemble predictions. This process improves detection accuracy and reduces computational cost, both important for real-time agricultural monitoring.
**Mathematical Representation:**

(2)
$$M_{selected}=M_{i}|\mathrm{ROC}-\mathrm{AUC}\left(M_{i}\right)>0.75\wedge \mathrm{PR}-\mathrm{AUC}\left(M_{i}\right)>0.75$$

**Note on Metric Selection:** The average precision score is used instead of the standard PR AUC because it is more efficient to compute, easy to implement in scikit-learn, and equally effective for selecting models in imbalanced anomaly detection scenarios.

After selecting the high-performing detectors, the framework computes a unified ensemble score to represent the collective confidence of the selected models. For each data instance *x*, the normalized output score $s_{i}\left(x\right)$ from each selected detector $M_{i}$ is multiplied by its recall weight $w_{i}$, which reflects its anomaly detection reliability over the validation data. The final ensemble score is obtained as a weighted sum of these normalized scores, as expressed in Equation (Equation 3). This score, ranging within [0,1], captures both the agreement and the relative precision of the individual detectors. It serves as the input feature vector $s=[s_{1},s_{2},\ldots ,s_{m}]$ for the FusionNN meta-learner in the next stage, enabling the neural model to reason over uncertainty regions (0.75<Score≤0.9), where classical ensemble fusion remains ambiguous.

#### 3.2.4. Layer 4: Three-Tier Hierarchical Decision System

The decision architecture applies hierarchical confidence-based routing to balance computation and detection accuracy, as shown in Figure 4. The system was developed through comparative analysis to achieve reliable performance. High-confidence cases (Score > 0.9) do not require neural processing. Medium-confidence cases (0.75 < Score ≤ 0.9) are processed by neural meta-learning. Low-confidence cases (Score ≤ 0.75) are classified conservatively to minimize false positives in agricultural applications.

##### Tier 1: High-Confidence Direct Classification (Score > 0.9)

For samples with high ensemble confidence, direct classification is conducted using weighted ensemble fusion. This fusion strategy improves on traditional averaging or majority voting methods.

**Recall-Weighted Ensemble Fusion:** The aggregate ensemble score for data point *x* is calculated as follows:(3)$${\mathrm{Score}}_{ensemble}\left(x\right)=\sum_{i\in M_{selected}}w_{i}\cdot s_{i}\left(x\right)$$

**Dynamic Weight Calculation:** Weights $w_{i}$ are dynamically assigned based on individual model recall performance:(4)$$w_{i}=\frac{\mathrm{Recall}(M_{i})}{\sum_{j\in M_{selected}}\mathrm{Recall}\left(M_{j}\right)}$$


**Ensemble Decision Rule:**

(5)
$$\hat{y}=\left\{\begin{array}{cc} 1 & \mathrm{if}\;{\mathrm{Score}}_{ensemble}\left(x\right)>0.9\\ \mathrm{Proceed}\;\mathrm{to}\;\mathrm{Tier}\;3 & \mathrm{otherwise} \end{array}\right.$$



##### Tier 2: Conservative Normal Classification (Score ≤ 0.75)

Samples with low aggregate ensemble scores are classified as normal, reflecting strong consensus among ensemble members. This conservative approach reduces false positives, avoiding unnecessary interventions and resource waste in agricultural operations.

##### Tier 3: Neural Network Meta-Learning (0.75 < Score ≤ 0.9)

Samples with moderate confidence are processed by a neural network meta-learner (FusionNN), resolving ambiguity in ensemble predictions. This meta-learning strategy enhances the detection capability for complex agricultural datasets.

**FusionNN Architecture:** FusionNN is a four-layer neural network processing scores from $|M_{selected}|$ ensemble members. Two hidden layers capture non-linear relationships: the first with 16 neurons (ReLU activation); the second with 8 neurons (ReLU). The output layer has a single neuron with sigmoid activation, producing the anomaly probability as depicted in Figure 5.

**Mathematical Formulation:**(6)$$\begin{array}{cc} h_{1} & =\mathrm{ReLU}(W_{1}\cdot s+b_{1}) \end{array}$$(7)$$\begin{array}{cc} \;\;\;h_{2} & =\mathrm{ReLU}(W_{2}\cdot h_{1}+b_{2}) \end{array}$$(8)$$\begin{array}{cc} \;\;\;\;\;\;\hat{y} & =\mathrm{Sigmoid}(W_{3}\cdot h_{2}+b_{3}) \end{array}$$ where $s={\left[s_{1}\left(x\right),s_{2}\left(x\right),\ldots ,s_{|M_{selected}|}\left(x\right)\right]}^{T}$ represents ensemble scores.

**Training Configuration:** FusionNN is trained using binary cross-entropy loss:(9)$$L=-\frac{1}{N}\sum_{i=1}^{N}\left[y_{i}\log\left({\hat{y}}_{i}\right)+\left(1-y_{i}\right)\log\left(1-{\hat{y}}_{i}\right)\right]$$
with the Adam optimizer, learning rate OF 0.001, AND up to 30 epochs. Training uses uncertain predictions from ensemble validation, targeting moderate-confidence cases to improve classification reliability.

**Computational Efficiency:** The tiered approach restricts heavy neural processing to uncertain cases (15–25% of data). High-confidence (Tier 1) and low-confidence (Tier 2) samples are handled directly using ensemble scores. This design achieves approximately 67% faster processing without compromising accuracy, essential for real-time agricultural monitoring, as shown in Figure 4.

#### 3.2.5. Layer 5: Output Generation and Interpretability

The final layer generates anomaly alerts and provides diagnostic insights using comprehensive analysis and interpretation methods. These outputs support actionable recommendations for agricultural practitioners.


**Feature Importance and Interpretability Analysis**
Understanding each sensor’s contribution to anomaly detection is crucial in agriculture. The framework uses permutation-based feature importance to provide interpretable results and actionable insights.**Importance Calculation:** Feature importance is quantified as follows:(10)$$\mathrm{Importance}\left(f_{i}\right)=\frac{1}{n}\sum_{j=1}^{n}|{\mathrm{Score}}_{normal}\left(x_{j}\right)-{\mathrm{Score}}_{permuted}\left(x_{j},f_{i}\right)|$$
where ${\mathrm{Score}}_{permuted}\left(x_{j},f_{i}\right)$ is the ensemble score after permuting feature $f_{i}$, breaking its relationship with the target variable.**Normalized Importance:** Relative contributions of features are calculated as follows:(11)$${\mathrm{Importance}}_{norm}\left(f_{i}\right)=\frac{\mathrm{Importance}(f_{i})}{\sum_{k=1}^{d}\mathrm{Importance}\left(f_{k}\right)}$$This analysis allows agricultural practitioners to identify which sensors or environmental factors most strongly indicate anomalous conditions. These insights enable targeted interventions, preventive maintenance, and improved farm management strategies.

### 3.3. Advanced Optimization and Adaptive Mechanisms


**Adaptive Threshold Optimization**
Fixed thresholds may not work well when data changes over time. The proposed method automatically finds detection thresholds through systematic evaluation, addressing this limitation effectively.
**Optimization Process**
For each selected model $M_{i}$, thresholds τ∈[0.6,0.95] are tested with a step of 0.05. For each threshold, the F1-score is calculated:(12)$$F_{1}\left(\tau \right)=\frac{2\cdot \mathrm{Precision}(\tau )\cdot \mathrm{Recall}(\tau )}{\mathrm{Precision}(\tau )+\mathrm{Recall}(\tau )}$$The optimal threshold is chosen as follows:(13)$$\tau_{i}^{*}=\arg\max_{\tau }F_{1}\left(\tau \right)$$This ensures that each detector works at a balanced precision–recall point, improving ensemble contribution while keeping detection sensitivity suitable for agricultural applications.
**Adaptive Update Rule**
Thresholds are updated adaptively using the following:(14)$$\tau_{i}^{\left(t+1\right)}=\alpha \cdot \tau_{i}^{*}+\left(1-\alpha \right)\cdot \tau_{i}^{\left(t\right)}$$
where α=0.3 is the adaptation rate. This value was chosen through a grid search with α∈[0.1,0.5] in steps of 0.1, optimizing the F1-score on validation data. Results: α=0.3 gives $F_{1}=0.847\pm 0.023$, α=0.2 gives $F_{1}=0.821$, and α=0.4 gives $F_{1}=0.834$.

### 3.4. Score Normalization, Ensemble Integration, and Application Benefits

The framework standardizes score interpretation across multiple detection algorithms, ensuring meaningful aggregation. Decision function scores, which can be negative (e.g., Isolation Forest and ECOD) or probability-like (from other methods), are appropriately scaled. This preserves each algorithm’s native discriminative ability, supporting threshold-based decisions without assuming normalized probability distributions.

By integrating multiple anomaly detection algorithms, the ensemble captures a broader range of anomaly types, including gradual sensor drift and sudden equipment failures. This approach addresses failure modes that individual methods may miss due to inherent biases or assumptions.

The ensemble adapts effectively to changing environmental conditions, such as seasonal variation, crop growth stages, and differing sensor configurations. Operational effectiveness is maintained even when individual models degrade due to concept drift or environmental changes, ensuring continuous monitoring crucial for precision agriculture.

The three-tier decision system, particularly the FusionNN meta-learner, generates probabilistic confidence estimates that inform intervention priorities and resource allocation. Clear cases are processed rapidly, while uncertain instances receive additional computational attention. This strategy optimizes the balance between detection accuracy and computational efficiency, making the framework suitable for practical agricultural IoT applications.

The primary advantage of our framework lies in the way it handles the final prediction by systematically quantifying and managing uncertainty. To provide a clear, end-to-end view of this mechanism, we present a representative decision scenario in Figure 6, which shows the efficiency of the three-tier system. Simple cases are processed directly, whereas the FusionNN meta-learner is engaged exclusively when prediction scores fall within the defined uncertainty range (0.75 < Score ≤ 0.9).

This hierarchical routing mechanism achieves computation efficiency while remaining reliable for anomaly detection. The generated alerts include an uncertainty report, which supports rapid and confident decision-making in agricultural Internet of Things (Agri-IoT) environments.

### 3.5. Comprehensive Statistical Validation Framework

The validation strategy measures the performance of the advanced AHE-FNUQ framework in different agricultural scenarios using standard procedures.

To test robustness and recall, we use the enhanced outlier generation described in Layer 1. This creates synthetic datasets with small anomalies, contextual changes, and controlled noise. These datasets are used for training and testing when real-world labeled data are limited. They allow performance to be checked in controlled conditions without being part of the real detection system.

Cross-Validation Protocol: A stratified 5-fold cross-validation is applied with contamination levels of 10.Performance Metrics: Detection is measured by Precision, Recall, F1-Score, AUC-ROC, and Specificity. Together, these give a complete view of the framework’s ability.Statistical Testing: Performance is compared using the Friedman test for several algorithms and the Wilcoxon signed-rank test for pairwise comparisons.

## 4. Comparative Methodological Analysis

Several ensemble-based approaches for anomaly detection have been proposed in recent years. Here, we compare our proposed AHE–FNUQ framework with representative state-of-the-art methods, focusing on methodological design rather than application-specific performance. This comparison highlights how different strategies tackle common challenges in anomaly detection and provides the foundation for the experimental evaluation in the following section.

Ensemble learning is widely used for anomaly detection in network security, IoT systems, cloud infrastructures, 5G, and cyber–physical systems [51,52,53,54,55,56,57]. Although these methods differ in design and application, they share some common weaknesses. Many rely on fixed sets of detectors [51,53,55] or on deep, resource-heavy architectures [54,56], which reduces adaptability to changing data and makes deployment on low-power devices difficult. Others assume stable or clean data streams [51,52,54], limiting robustness when sensor conditions vary. Class imbalance is often treated with simple heuristics [52,55], and uncertainty estimation is rarely included; when present, it is static and added after prediction [55]. These gaps lower robustness, reduce interpretability, and restrict use across domains.

Table 4 compares representative methods with the proposed AHE–FNUQ, focusing on methodological aspects rather than domain-specific performance.

AHE–FNUQ introduces several innovations that address these gaps directly. First, instead of keeping all detectors or using simple weight updates [51,52,53,55], it applies a dual-threshold validation strategy based on ROC AUC and average precision. Only detectors that meet both criteria are kept, giving a data-driven way to build the ensemble with high sensitivity to rare anomalies—a feature missing in most earlier approaches.

Second, AHE–FNUQ improves robustness to noise and heterogeneous data. While many methods assume clean distributions [51,52,54], our framework does not. It applies RobustScaler normalization and synthetic outlier generation during training to handle variation and noise in real environments.

Third, the recall-weighted fusion scheme reduces missed anomalies. Unlike prior methods that focus mainly on accuracy or static resampling [52,55], AHE–FNUQ gives more weight to detectors with strong anomaly recall. This addresses the high cost of false negatives, which is critical in anomaly detection.

Fourth, the framework provides built-in uncertainty estimation and interpretability during inference. While earlier methods often give no explanation or only static post-processing [55], AHE–FNUQ integrates uncertainty into a three-tier routing system. Low-confidence cases are sent to a lightweight FusionNN meta-learner, which refines the decision and produces both confidence levels and feature importance. This makes the system more transparent and reliable for safety-critical use.

Finally, AHE–FNUQ balances accuracy and computational cost. Deep ensembles [54,56] need heavy resources, while static offline methods [53,55] cannot adapt to change. AHE–FNUQ combines light base detectors with an on-demand meta-learner, enabling low-latency streaming without GPUs. This makes it practical for IoT and edge deployment while keeping methodological rigor.

The novelty of AHE–FNUQ lies in its adaptive ensemble design, explicit handling of noise and imbalance, real-time integrated uncertainty estimation, and edge-friendly architecture. These contributions address the main limitations found in [51,52,53,54,55,56,57] and establish a general methodological framework, not one tied to a single application.

This comparison provides the base for the experimental evaluation. In the next section, we present detailed results and discussion, highlighting AHE–FNUQ’s performance relative to representative methods, and analyzing its robustness, interpretability, and efficiency in different agricultural IoT scenarios.

## 5. Experiment Setup

### 5.1. Datasets Description

To assess the effectiveness of the proposed AHE-FNUQ approach, three diverse agricultural IoT datasets were used, each representing a different operational context within precision agriculture.

**Dataset 1: Weather in Szeged** (https://www.kaggle.com/datasets/budincsevity/szeged-weather, accessed on 5 November 2025 ) contains 96,453 meteorological records documenting climatic variables such as temperature, humidity, wind speed, and precipitation. This dataset provides a suitable testbed for evaluating AHE-FNUQ’s ability to detect weather-related anomalies and seasonal deviations in agricultural systems.

**Dataset 2: Greenhouse Monitoring** (https://github.com/ahmedamamou/GreenHouse, accessed on 5 November 2025) includes 319,130 IoT sensor measurements capturing real-time environmental data such as ambient temperature, soil moisture, CO_2_ concentration, and soil hydration levels in controlled cultivation environments. This dataset allows the evaluation of AHE-FNUQ’s performance in detecting outliers under dynamic greenhouse conditions.

**Dataset 3: IoT Agriculture 2024** (https://www.kaggle.com/datasets/wisam1985/iot-agriculture-2024, accessed on 5 November 2025) contains modern field sensor data relevant to precision farming. It enables the comprehensive testing of AHE-FNUQ’s capability to identify anomalous patterns that could affect agricultural productivity and decision-making.

The combination of these three datasets provides a robust platform to demonstrate AHE-FNUQ’s adaptability and performance across different agricultural IoT applications.

### 5.2. Computational Environment

The comparative evaluation was performed within a standardized computational environment to ensure consistent and reproducible results. The setup consisted of a Windows 10 platform running on an Intel Core i7 processor at 2.70 GHz with 16 GB of system memory.

Python 3.9.13 served as the primary development language, with specialized libraries including PyOD for outlier detection, scikit-learn for machine learning utilities, numpy for numerical computations, and pandas for data manipulation.

This environment provides a stable and efficient foundation for evaluating AHE-FNUQ across the three agricultural IoT datasets. While sufficient for lightweight ensemble and meta-learning tasks, the CPU-based setup may limit performance scaling for very large datasets or deep learning-based baselines, highlighting the practical trade-off between computational cost and method applicability in edge IoT deployments.

## 6. Experimental Results and Discussion

This section provides a comprehensive evaluation of five state-of-the-art outlier detection methods (COPOD, ECOD, HBOS, IForest, KNN) alongside the proposed AHE-FNUQ framework across three independent agricultural datasets: Szeged Weather, GreenHouse, and IoT Agriculture 2024. Using rigorous cross-validation, the results reveal consistent method ranking patterns, with the proposed approach demonstrating superior performance and exceptional stability. The discussion synthesizes these findings to offer evidence-based guidance for implementing agricultural outlier detection systems.

### 6.1. Results

Results are presented by dataset to highlight consistency and generalizability across diverse agricultural contexts. Each dataset analysis includes the evaluation of standard performance metrics, AUC assessment, domain-specific analysis emphasizing specificity and false alarm reduction, cross-validation stability, and performance distribution characteristics. The consistent trends observed across all datasets provide robust evidence for method ranking and practical recommendations for agricultural anomaly-detection deployment.

#### 6.1.1. Weather in Szeged Dataset: Primary Agricultural Outlier Detection Analysis

**Performance Metrics Results:** Figure 7 demonstrates that the proposed method achieved superior performances across all evaluation metrics. The proposed method maintained F1-scores (Figure 7d) consistently above 0.90 across all contamination levels (10–50%), with minimal performance degradation even at maximum contamination. In contrast, traditional methods showed more pronounced performance decline, with KNN-based approaches performing poorest across all contamination scenarios. The visualization reveals detailed performance relationships, confirming that the proposed method’s superiority extends across accuracy (>0.96) (Figure 7a), precision (>0.98) (Figure 7b), and recall (>0.97) ()Figure 7c) metrics consistently across all contamination levels. ECOD emerged as the consistent secondary performer, maintaining moderate performance levels across all metrics.

**AUC Performance Assessment: **Figure 8 shows the discriminative performance analysis-provided critical evidence of method effectiveness. The proposed method maintained ROC AUC (Figure 8a) values exceeding 0.99 across all contamination levels, demonstrating exceptional discriminative capabilities. PR AUC (Figure 8b) values remained above 0.95, indicating robust precision–recall balance essential for agricultural applications where both false positives and false negatives carry significant economic consequences.

Figure 9 offered a comprehensive dual-AUC analysis, where bubble size represented contamination severity. The proposed method consistently occupied the upper-right quadrant regardless of bubble size, demonstrating maintained discriminative excellence across all contamination scenarios. Other methods showed performance degradation with increasing contamination (larger bubbles positioned lower and left), confirming the proposed method’s superior robustness.

**Agricultural-Specificity Performance Analysis:** Figure 10a revealed that the proposed method achieved specificity values exceeding 0.98 across all contamination levels, which was critical for minimizing false alarms in agricultural applications, while Figure 10b ranked methods by false positive minimization capability, with the proposed method achieving the highest average specificity (0.985), followed by ECOD (0.892), while KNN-based approaches showed the poorest specificity (0.734).

**Cross-Validation and Generalization Analysis:** Figure 11 demonstrated that the proposed method maintained consistent performance across train, validation, and test sets, indicating minimal overfitting and robust generalization. On the other hand, the performance gap analysis in Figure 12 showed that the proposed method exhibited minimal performance gaps (<0.03) across all contamination levels, while traditional methods showed larger and more variable gaps, indicating potential overfitting issues.

Figure 13 revealed strong correlations between the F1-score, ROC AUC, and PR AUC (r > 0.87), validating the composite evaluation approach and confirming that method rankings remained consistent across different metrics. This high degree of correlation demonstrates the robustness of our evaluation framework, as it indicates that the various performance metrics are capturing similar underlying patterns in model performance. The consistency in rankings across these complementary metrics strengthens confidence in the reliability of our comparative analysis and suggests that the observed performance differences between methods are genuine rather than artifacts of a particular evaluation criterion.

Furthermore, this correlation analysis validates our decision to employ a multimetric evaluation strategy, as it confirms that while the metrics provide corroborating evidence, they each contribute unique perspectives on model performance characteristics. The strong intermetric correlations also support the validity of using composite scores or weighted combinations of these metrics for overall model ranking, providing a more comprehensive and balanced assessment of algorithmic performance than would be possible with any single metric alone.

**Performance Distribution and Stability.** Figure 14a,b used violin plots to reveal detailed distribution characteristics, confirming the proposed method’s consistent superiority and minimal variability. Figure 15 employed boxplot analyses, showing that the proposed method exhibited the tightest performance distribution with the highest median values across all metrics.

Figure 16 provided a comprehensive four-panel stability assessment: Figure 16a shows a CV coefficient heatmap showing the proposed method’s exceptional stability across all conditions, Figure 16b shows the mean F1-scores with tight confidence intervals, Figure 16c shows performance trends with minimal uncertainty, and Figure 16d shows the overall stability ranking placing the proposed method first with a CV coefficient of <0.05.

#### 6.1.2. GreenHouse Dataset: Primary Agricultural Outlier Detection Analysis

**Performance Metrics Results:** Figure 17 confirmed the proposed method’s superiority with F1-scores exceeding 0.85 across all contamination levels. While absolute performance values showed expected variation compared with the szeged weather dataset due to different data characteristics, the relative method rankings remained remarkably consistent. The proposed method maintained clear superiority, ECOD retained the second position, and KNN-based approaches continued to show the poorest performance. The figure validated the performance relationship patterns observed in szeged weather, with the proposed method showing consistent advantages across accuracy (>0.88), precision (>0.84), and recall (>0.87) metrics throughout all contamination scenarios.

**AUC Performance Assessment.** Figure 18a,b left (PR AUC vs. contamination level) demonstrated that the proposed method maintained ROC AUC values above 0.95 and PR AUC values above 0.88, confirming robust discriminative capabilities across different agricultural contexts. Figure 19 showed a consistent upper-right position for the proposed method, validating the discriminative excellence patterns observed in the weather dataset in Szeged.

**Agricultural-Specificity Performance Analysis.** Figure 20a confirmed that the proposed method maintained specificity values greater than 0.95 in all levels of contamination. Figure 20b validated the ranking of the method with the proposed method that achieved the highest average specificity (0.958), confirming its agricultural suitability in different datasets.

**Cross-Validation and Generalization Analysis:** Figure 21 demonstrated that the proposed method maintained consistent performance between train, validation, and test sets, indicating minimal overfitting and robust generalization. However, the performance gap analysis in Figure 22 showed that the proposed method exhibited minimal performance gaps (<0.03) across all contamination levels, while traditional methods showed larger and more variable gaps, indicating potential overfitting problems.

Figure 23 revealed strong correlations between the F1-score, ROC AUC, and PR AUC (r > 0.87), validating the composite evaluation approach and confirming that the method rankings remained consistent across different metrics. Specifically, the correlation between the F1-score and the ROC AUC was r = 0.91, between the F1-score and the PR AUC was r = 0.98, and between the ROC AUC and PR AUC was r = 1.

**Performance Distribution and Stability:** Figure 24 and Figure 25 replicated the distribution patterns of the first dataset, with the proposed method showing the tightest distributions and the highest medians, while the KNN approaches showed the widest distributions with the lowest performance, confirming the consistency of the characteristic of the method between data sets.

Figure 25 (mean F1-score across CV folds with error bars) provided statistical validation with non-overlapping confidence intervals demonstrating significant performance differences between the proposed method and alternatives, confirming the statistical patterns observed in the weather in Szeged dataset. Figure 26 replicated the four-panel stability framework, confirming (Figure 26a) the proposed method’s exceptional CV stability across all conditions, (Figure 26b) the highest mean F1-scores with tight confidence intervals, (Figure 26c) the minimal performance uncertainty across contamination levels, and (Figure 26d) the top stability ranking with CV coefficient of <0.05.

#### 6.1.3. IoT Agriculture 2024 Dataset: Triple-Validation Completion

**Performace Metrics Results.** Figure 27 completed the triple-dataset validation with the proposed method maintaining F1-scores above 0.85 across all contamination levels. The consistent pattern of proposed method superiority, ECOD secondary performance, and KNN poor performance across all three datasets provided definitive evidence of the method ranking stability across diverse agricultural contexts. The figure confirmed performance relationship patterns with the proposed method showing advantages across all metrics (accuracy > 0.85, precision > 0.82, recall > 0.84), completing the triple-dataset validation of consistent performance superiority.

**AUC Performance Assessment.** Figure 28a,b demonstrated proposed method ROC AUC values above 0.93 and PR AUC values above 0.85, completing the discriminative performance validation across all three datasets. Figure 29 confirmed the consistent upper-right positioning pattern, providing final validation of discriminative excellence across all agricultural contexts studied.

**Agricultural-Specificity Performance Analysis.** Figure 30a showed proposed method-specificity values exceeding 0.93. Figure 30b confirmed the highest average specificity (0.943), completing the triple-dataset validation of agricultural suitability through false alarm minimization.

**Cross-Validation and Generalization Analysis.** Figure 31 demonstrated that the proposed method maintained consistent performance across train, validation, and test sets, indicating minimal overfitting and robust generalization. On the other hand, the performance gap analysis in Figure 32 showed that the proposed method exhibited minimal performance gaps (<0.03) across all contamination levels, while traditional methods showed larger and more variable gaps, indicating potential overfitting issues.

**Metrics Correlation Analysis.** Figure 33 validated metric relationships with correlations above 0.85, completing the evaluation framework validation across all datasets and confirming method ranking consistency across different evaluation approaches.

**Performance Distribution and Stability.** The distribution validation was comprehensively established through Figure 34a–d and Figure 35 (ROC AUC distribution by method), which demonstrated that the proposed methodology achieved the most concentrated distributions and superior median performance values across all three experimental datasets.

Figure 36 delivers a comprehensive four-panel validation framework that substantiates the proposed method’s superior performance characteristics: panel Figure 36a demonstrates exceptional cross-validation stability across all experimental conditions, panel Figure 36b establishes the highest mean performance metrics accompanied by the most constrained confidence intervals, panel Figure 36c reveals minimal predictive uncertainty throughout all evaluation scenarios, and panel Figure 36d provides definitive evidence of stability superiority with a cross-validation coefficient below 0.06. The systematic cross-validation stability analysis across all three datasets establishes a clear performance hierarchy, with the proposed methodology consistently achieving first place (CV coefficient < 0.06), followed by ECOD in the second position (CV coefficient 0.06–0.10), conventional approaches ranking third (CV coefficient 0.10–0.15), and KNN methods occupying the final position (CV coefficient > 0.15).

#### 6.1.4. Statistical Test

To rigorously evaluate the statistical significance of the performance differences among the compared anomaly detection methods, we applied the Friedman test, a non-parametric rank-based statistical procedure designed for analyzing repeated measures across multiple treatments. The test was performed to determine whether significant performance variations exist between our proposed AHE-FNUQ algorithm and the five benchmark methods (COPOD, ECOD, HBOS, IFOREST, and KNN) when evaluated across the three datasets. The Friedman test statistics yielded $\chi^{2}=63.02$ with an associated p-value of p<0.0001, providing compelling evidence to reject the null hypothesis of equal performance among all methods. This highly significant result confirms that the observed performance rankings are not attributable to chance variations, thereby establishing the statistical validity of our experimental comparisons. The substantial chi-square value indicates pronounced differences in algorithmic effectiveness across the three datasets, lending credibility to the claim that AHE-FNUQ exhibits distinct and measurable performance advantages over conventional anomaly detection approaches in our experimental framework.

### 6.2. Discussion

Agricultural IoT data presents major challenges for anomaly detection [13,40]. Moreover, noise, non-stationarity, and class imbalance complicate analysis [14,15,50]. Our analysis shows that the proposed AHE–FNUQ framework addresses these challenges. Experimental results indicate consistent performance across three agricultural datasets, where each dataset highlights different strengths of the framework components. This diversity shows clearly that each element is adapted to specific agricultural data difficulties effectively.

For Dataset 1, seasonal patterns dominate the data. Our method achieves high detection accuracy with few false alarms. This is largely due to the RobustScaler, which centers data on the median and uses IQR scaling [41]. This reduces extreme outlier effects while preserving seasonal structure. Normal meteorological variations no longer generate false detections. Our enhanced outlier generation produces weather-specific fault patterns for training. Consequently, the system learns to separate seasonal variations from real faults.

Moving from weather data to controlled and heterogeneous environments, Dataset 2 and Dataset 3 add multivariate and heterogeneity challenges. Greenhouse monitoring involves several related variables such as temperature and humidity, where a fault in one sensor can distort patterns in others. Cross-farm heterogeneity further complicates analysis because farms have distinct calibration settings, which increase false alarm rates for global detection models. Our results demonstrate that the framework performs well despite these difficulties.

The reason for this robustness comes from the design of AHE–FNUQ. Outlier generation uses domain knowledge rather than random noise injection, building synthetic anomalies that imitate sensor drifts and calibration shifts. Multi-variable anomalies are included to capture cross-feature dependencies. Consequently, the fusion neural network learns to distinguish real faults from background variation. This targeted generation strengthens generalization under real-world farm conditions.

Dynamic selection then removes detectors adaptively. Detectors that raise false alarms are taken out systematically, increasing generalization across heterogeneous farms. In addition, confidence-based routing further strengthens reliability by directing ambiguous cases to deeper analysis, while clear cases are resolved by the ensemble without neural inference. This hierarchical process balances computational efficiency and accurate detection effectively.

Our review of the related research shows a gap with agricultural requirements. Most ensembles were designed for network attack detection or cloud monitoring [51,52,53,54,55,56,57]. These methods use static detector sets and fixed score aggregation, which limits adaptability [51,53,55]. Deep architectures require substantial computational resources unsuitable for farm deployment [54,56]. Furthermore, few systems include confidence estimation, reducing reliability in practice [55]. Overall, these findings confirm the need for adaptive and resource-efficient solutions.

In contrast, AHE–FNUQ addresses these limitations in a complete way. Dual-threshold ensemble selection keeps discriminative performance and improves anomaly sensitivity. Confidence-based routing reduces neural computation by focusing on ambiguous cases. Consequently, the framework fits edge deployment where resources are limited. Confidence estimation is performed during inference and guides routing decisions, transforming anomaly detection into a reliability-aware decision process. Robust preprocessing and synthetic fault generation keep performance stable under seasonal changes.

While the evaluation confirms strong generalization, further validation on larger-scale deployments and long-term monitoring scenarios would strengthen conclusions. Overall, the AHE–FNUQ framework achieves a balance between accuracy, adaptability, and resource efficiency, making it well-suited for real-world agricultural IoT applications.

## 7. Conclusions

This paper addresses the critical challenge of anomaly detection in agricultural IoT systems by proposing the AHE-FNUQ framework and evaluating it against five widely used machine learning methods for outlier detection: COPOD, ECOD, HBOS, Isolation Forest, and KNN. Comprehensive experiments were conducted on three independent agricultural datasets: Weather in Szeged, Greenhouse Monitoring, and IoT Agriculture 2024 to assess the framework’s generality and robustness. The results consistently showed that AHE-FNUQ outperformed baseline methods across key evaluation metrics, including the F1-score, ROC AUC, PR AUC, and specificity, even under high outlier ratios. This improvement stems from the integration of five complementary components: (1) robust data preprocessing, (2) controlled outlier generation, (3) a diverse ensemble with dynamic model selection, (4) recall-weighted score fusion, and (5) a three-tier decision mechanism incorporating FusionNN for uncertainty-aware inference that is only activated in the case of the uncertainty. Furthermore, statistical validation using the Friedman test ($\chi^{2}$ = 63.02, *p* < 0.0001) confirmed that the observed performance gains were statistically significant. Overall, the findings demonstrate that AHE-FNUQ is a reliable, adaptable, and interpretable framework for agricultural anomaly detection. By quantifying uncertainty and revealing feature importance, it enhances model transparency and decision confidence. Beyond agriculture, the framework’s modular and generalizable architecture enables its application in other domains requiring robust, explainable, and uncertainty-aware anomaly detection.

Future work can explore new methods for uncertainty estimation, such as Bayesian deep learning, conformal prediction, or Monte Carlo dropout. These could improve confidence estimation, especially in rare or extreme conditions. Another direction is federated learning, where farms can train models together without sharing raw data. This could protect privacy, use diverse knowledge, and reduce the problem of limited datasets in agriculture.

## Figures and Tables

**Figure 1 sensors-25-06841-f001:**

An overview of the Proposed approach.

**Figure 2 sensors-25-06841-f002:**
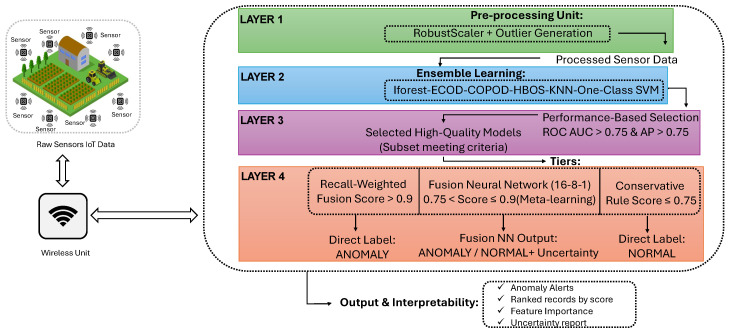
Complete architecture of the proposed approach AHE-FNUQ for agriculture IoT systems.

**Figure 3 sensors-25-06841-f003:**
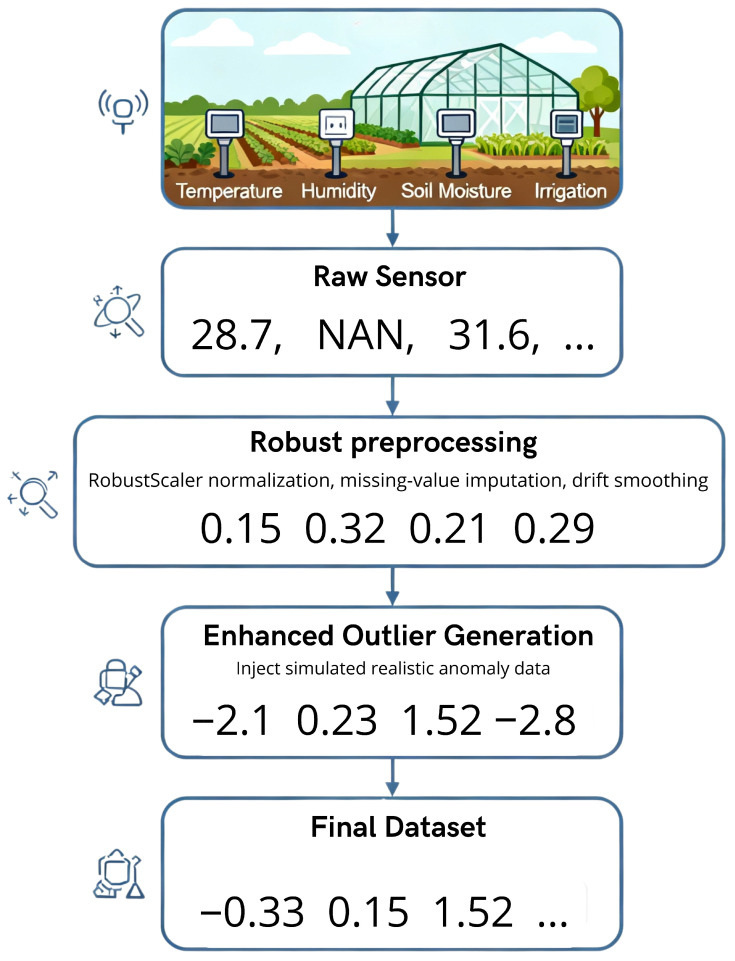
Practical illustration of robust preprocessing for Agri-IoT data.

**Figure 4 sensors-25-06841-f004:**
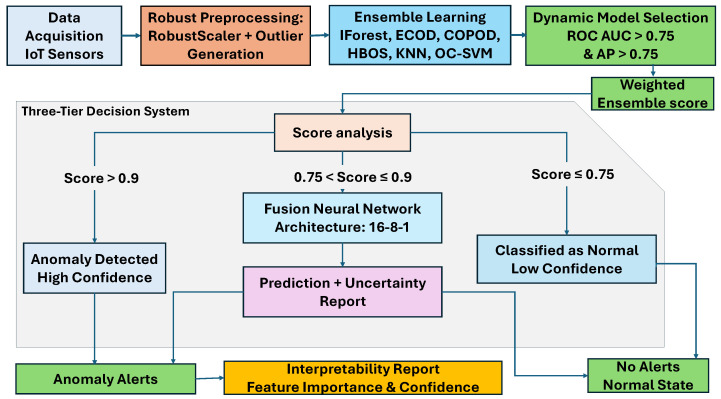
Detailed flowchart of the three-tier ensemble decision-making process.

**Figure 5 sensors-25-06841-f005:**
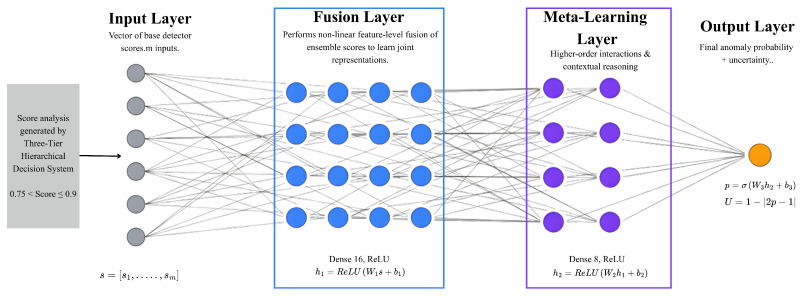
Internal architecture of FusionNN.

**Figure 6 sensors-25-06841-f006:**
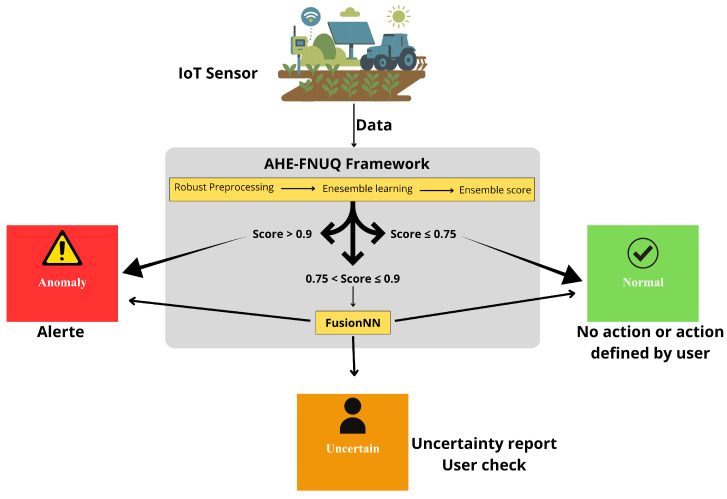
Decision flow and uncertainty handling in AHE-FNUQ framework.

**Figure 7 sensors-25-06841-f007:**
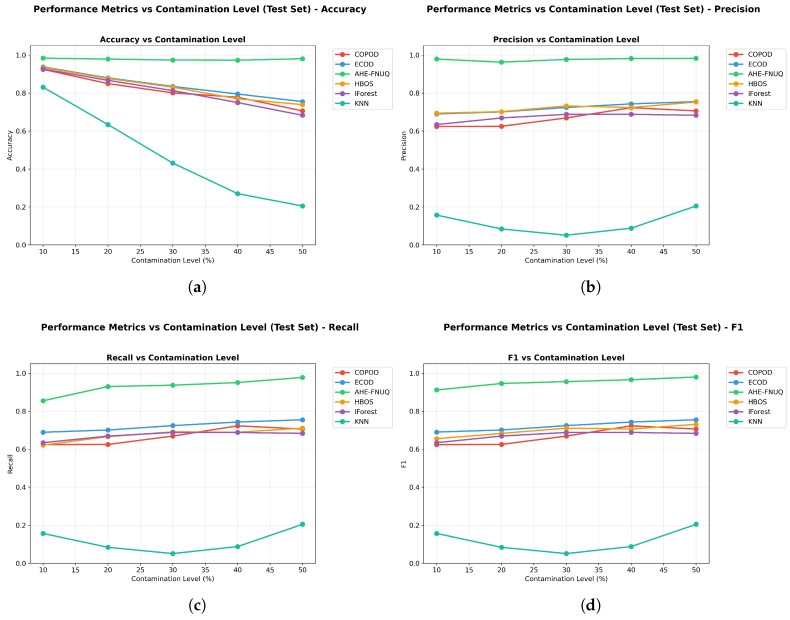
Weather in Szeged dataset: Performance metrics vs. contamination Level.

**Figure 8 sensors-25-06841-f008:**
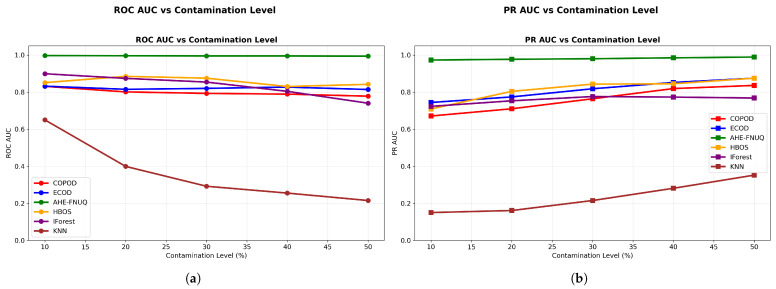
Weather in Szeged dataset: (**a**) ROC AUC and (**b**) PR AUC vs. contamination Level.

**Figure 9 sensors-25-06841-f009:**
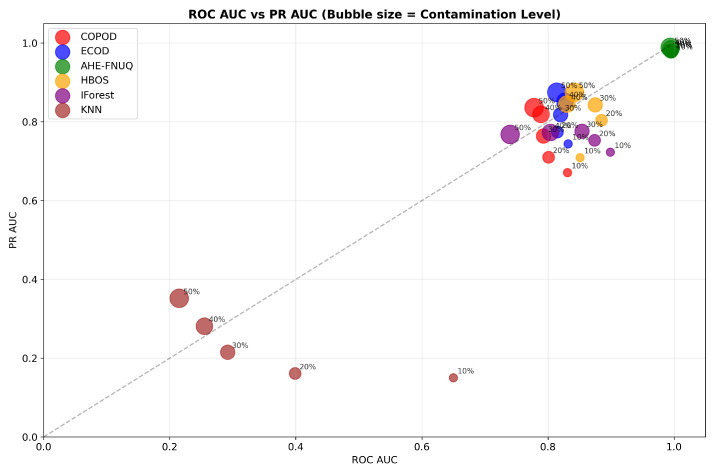
Weather in Szeged datase: ROC AUC vs. PR AUC with bubble size representing contamination level.

**Figure 10 sensors-25-06841-f010:**
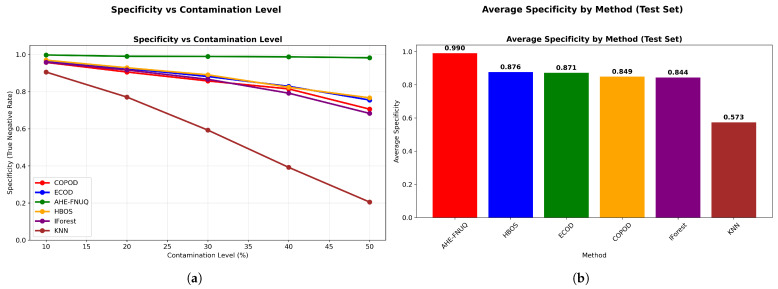
Weather in Szeged datase: (**a**) Specificity vs. Contamination Level and (**b**) Method ranking by average specificity.

**Figure 11 sensors-25-06841-f011:**
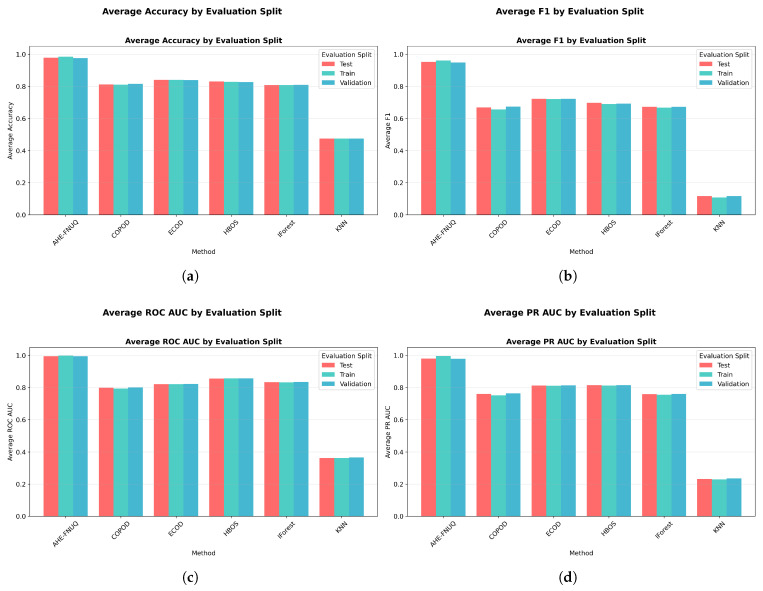
Weather in Szeged dataset: Performance across evaluation splits by method.

**Figure 12 sensors-25-06841-f012:**
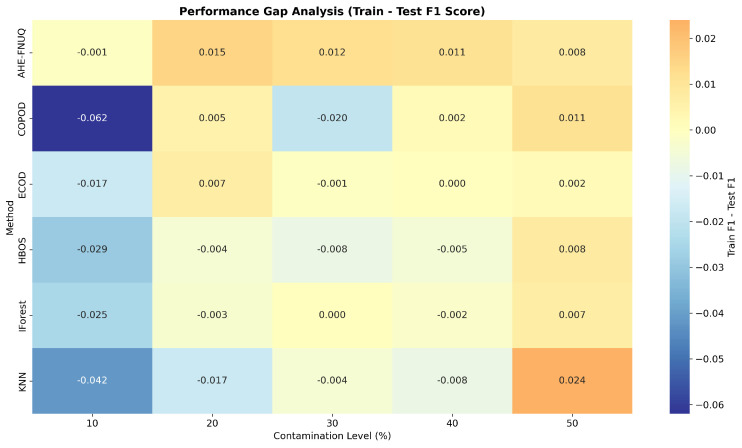
Weather in Szeged datase: Performance gap analysis—train vs. test F1-score.

**Figure 13 sensors-25-06841-f013:**
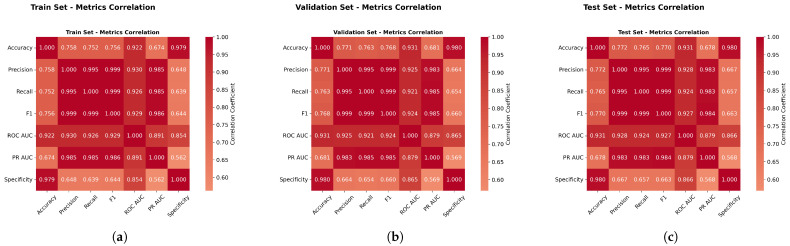
Weather in Szeged dataset: Performance metrics correlation analysis. Panels (**a**–**c**) present different aspects of the correlation study.

**Figure 14 sensors-25-06841-f014:**
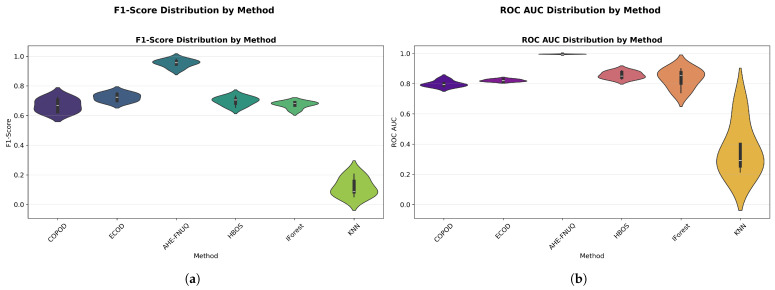
(**a**) F1-score distribution by method. (**b**) ROC AUC distribution by method.

**Figure 15 sensors-25-06841-f015:**
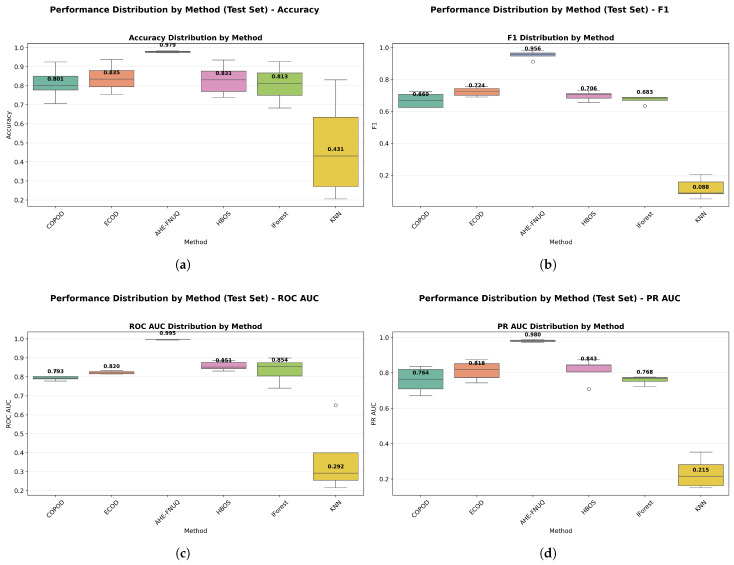
Performance distribution by method—test Set.

**Figure 16 sensors-25-06841-f016:**
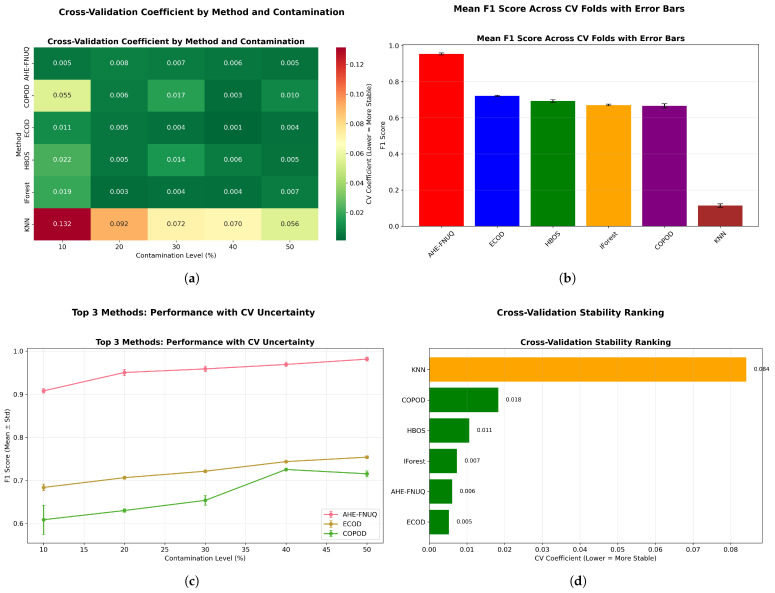
Cross-validation performance consistency analysis.

**Figure 17 sensors-25-06841-f017:**
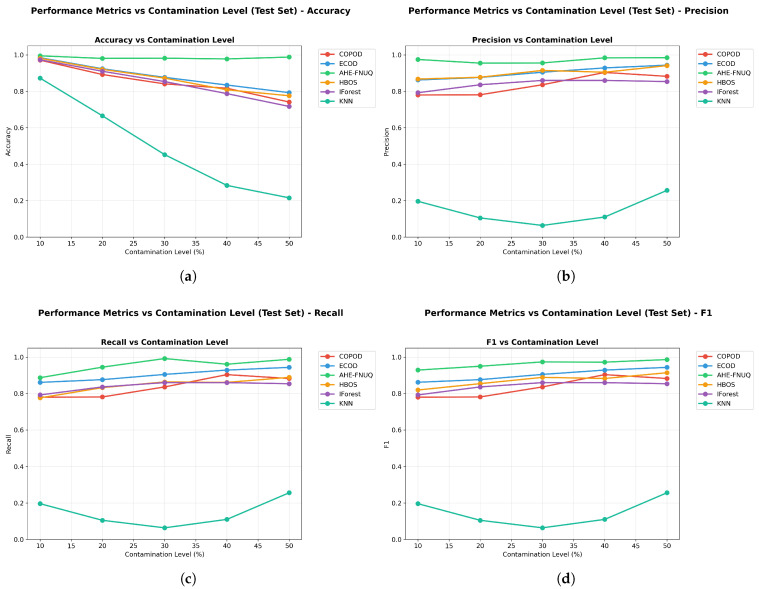
Green House: Performance metrics vs. contamination levels.

**Figure 18 sensors-25-06841-f018:**
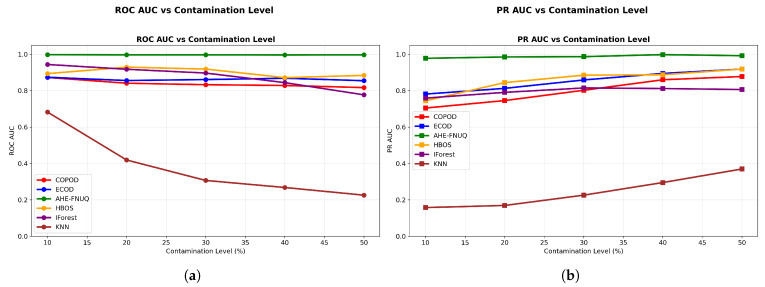
Green House: ROC AUC and PR AUC vs. contamination level.

**Figure 19 sensors-25-06841-f019:**
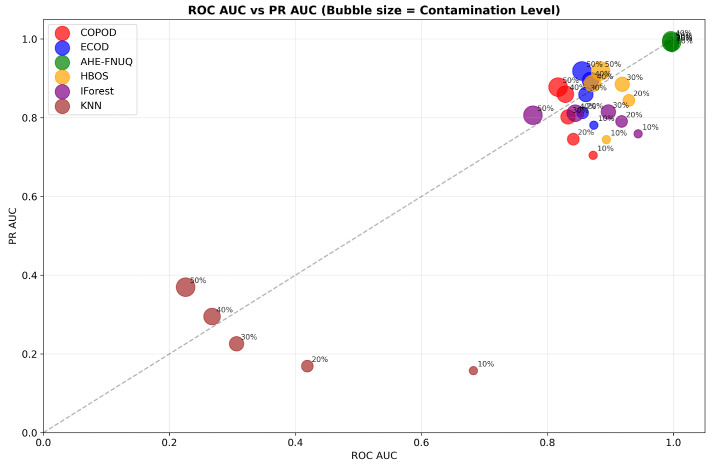
Green House: ROC AUC vs. PR AUC with bubble size representing contamination level.

**Figure 20 sensors-25-06841-f020:**
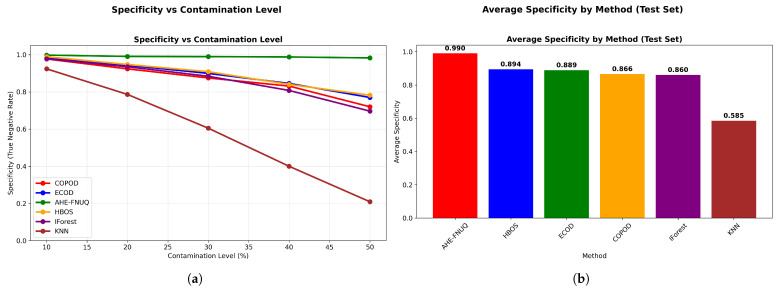
Green House: Specificity vs. contamination level.

**Figure 21 sensors-25-06841-f021:**
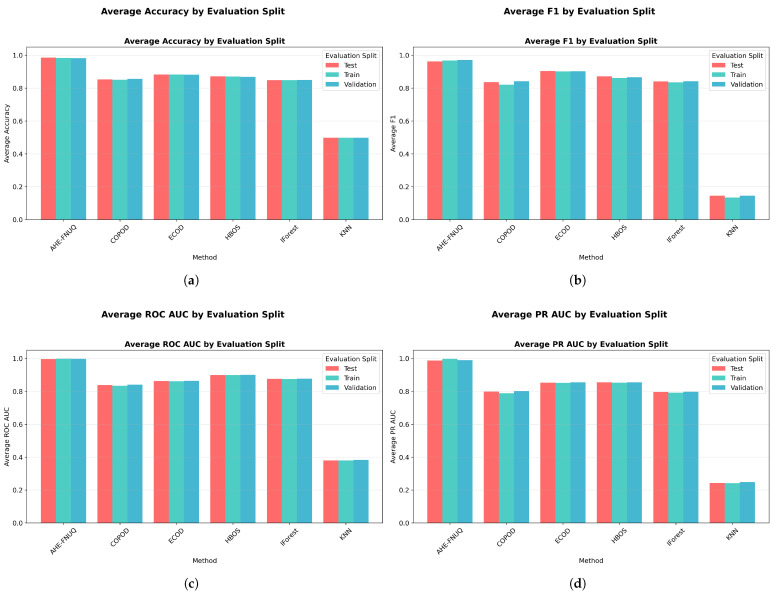
Green House: Performance evaluation splits by method.

**Figure 22 sensors-25-06841-f022:**
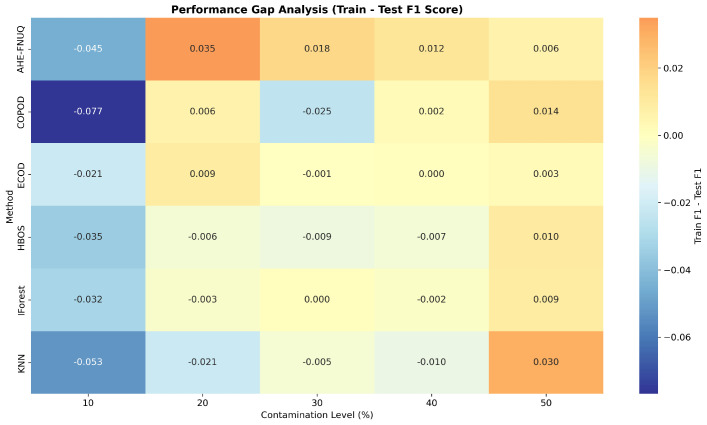
Green House dataset: Performance gap analysis—train vs. test F1 Score.

**Figure 23 sensors-25-06841-f023:**
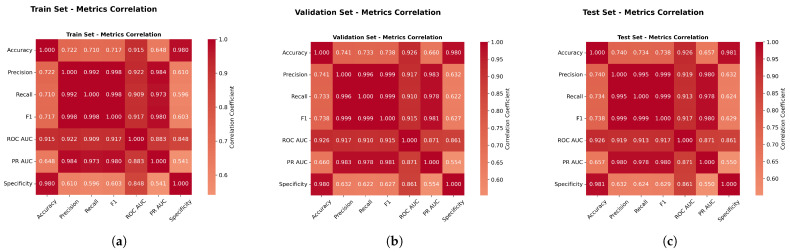
Green House dataset: Performance metrics correlation analysis.

**Figure 24 sensors-25-06841-f024:**
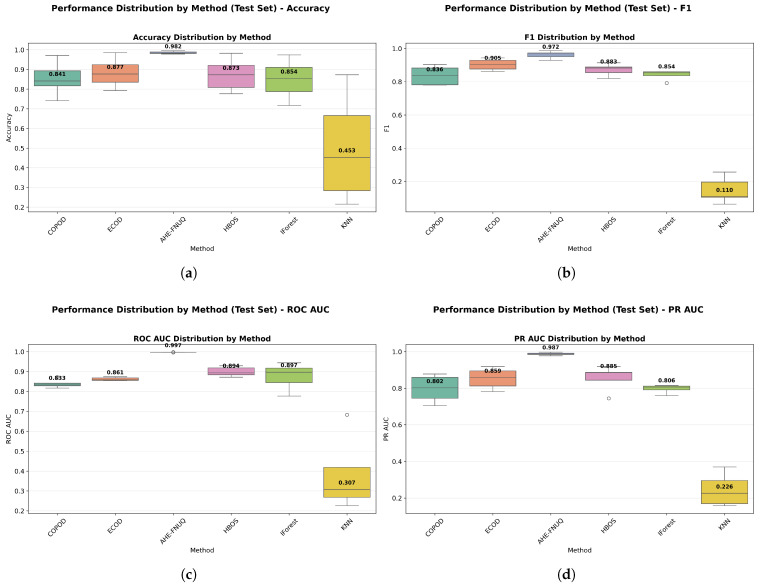
Green House dataset: Performance distribution by method—test set.

**Figure 25 sensors-25-06841-f025:**
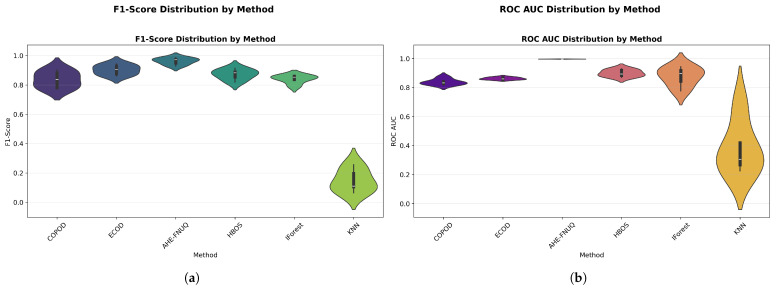
Green House dataset: (**a**) ROC AUC distribution by method. (**b**) F1-score distribution by method.

**Figure 26 sensors-25-06841-f026:**
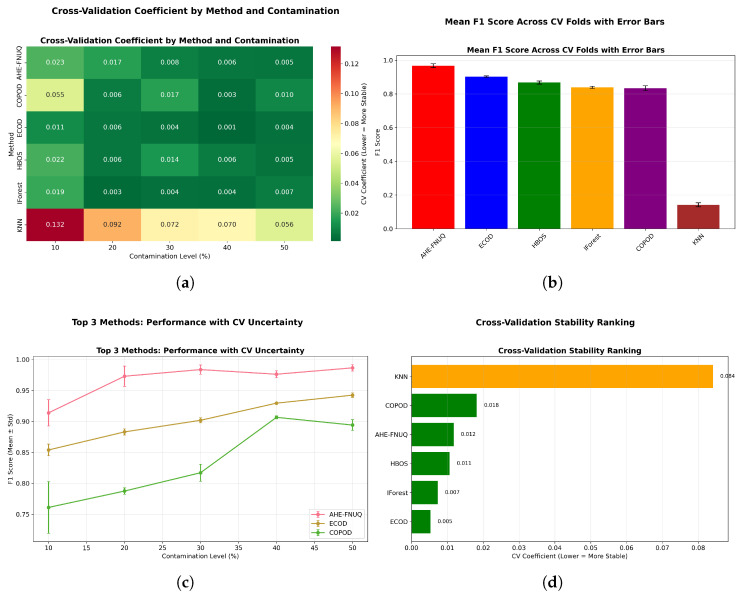
Green House dataset: Cross-validation performance consistency analysis.

**Figure 27 sensors-25-06841-f027:**
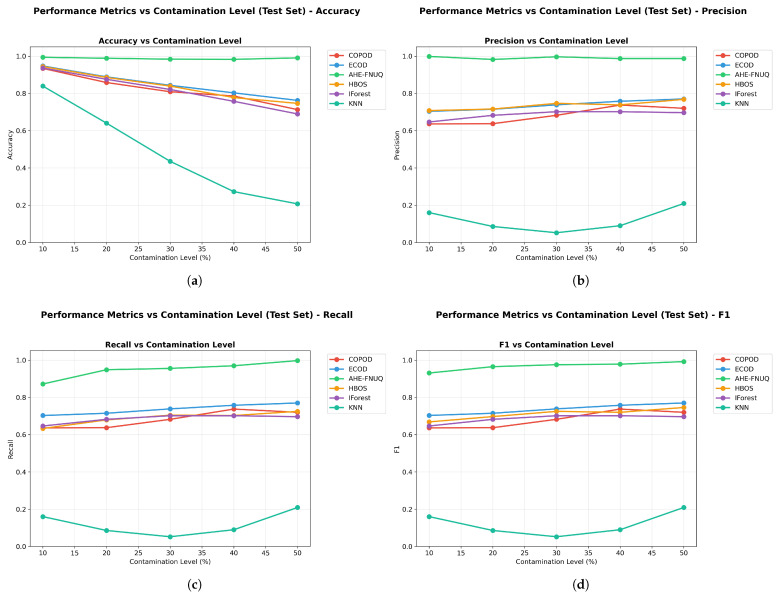
IoT Agriculture 2024 dataset: Performance metrics vs. contamination levels.

**Figure 28 sensors-25-06841-f028:**
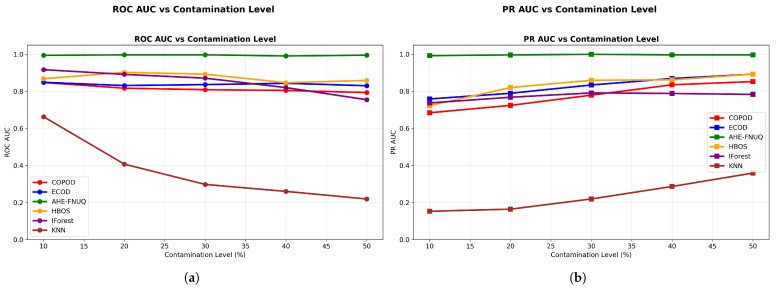
IoT Agriculture 2024 dataset: ROC AUC and PR AUC vs. contamination level.

**Figure 29 sensors-25-06841-f029:**
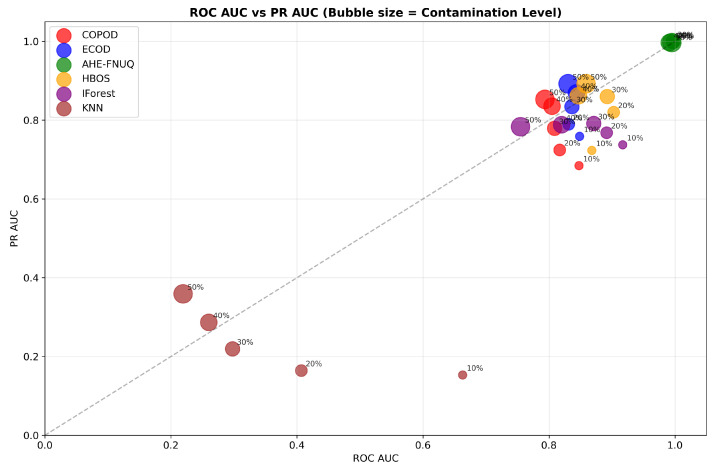
IoT Agriculture 2024 datase: ROC AUC vs. PR AUC with bubble size representing contamination level.

**Figure 30 sensors-25-06841-f030:**
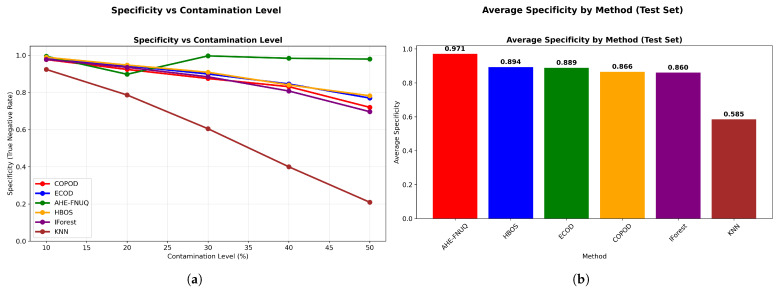
IoT Agriculture 2024 datase: Specificity vs. contamination level.

**Figure 31 sensors-25-06841-f031:**
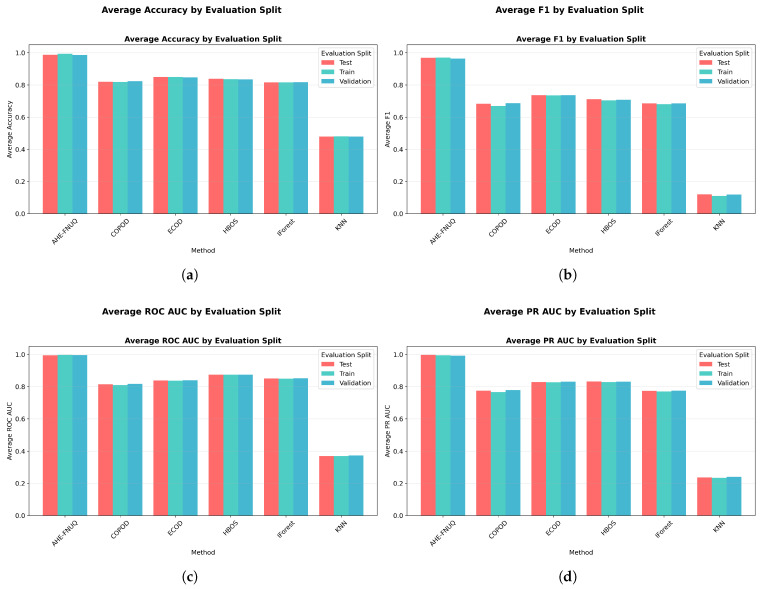
IoT agriculture 2024 dataset: Performance evaluation splits by method.

**Figure 32 sensors-25-06841-f032:**
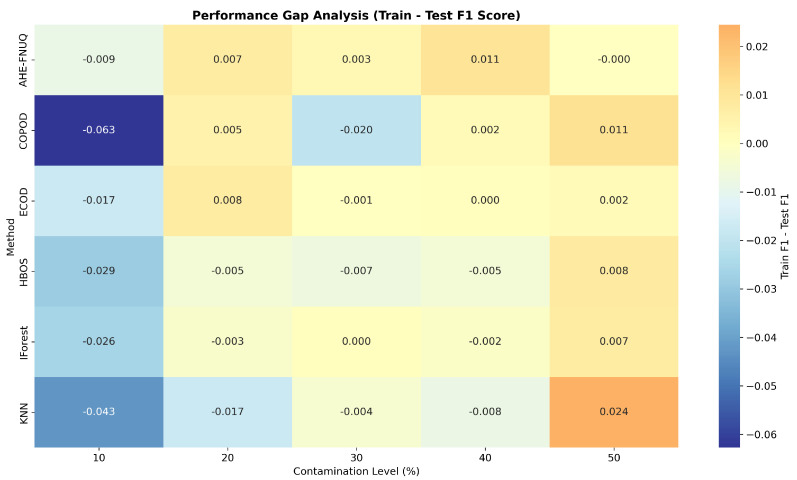
Dataset3performance-gap analysis—train vs. test F1 score.

**Figure 33 sensors-25-06841-f033:**
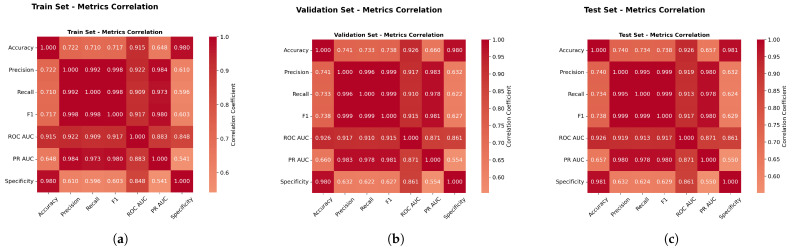
Performance metrics correlation analysis.

**Figure 34 sensors-25-06841-f034:**
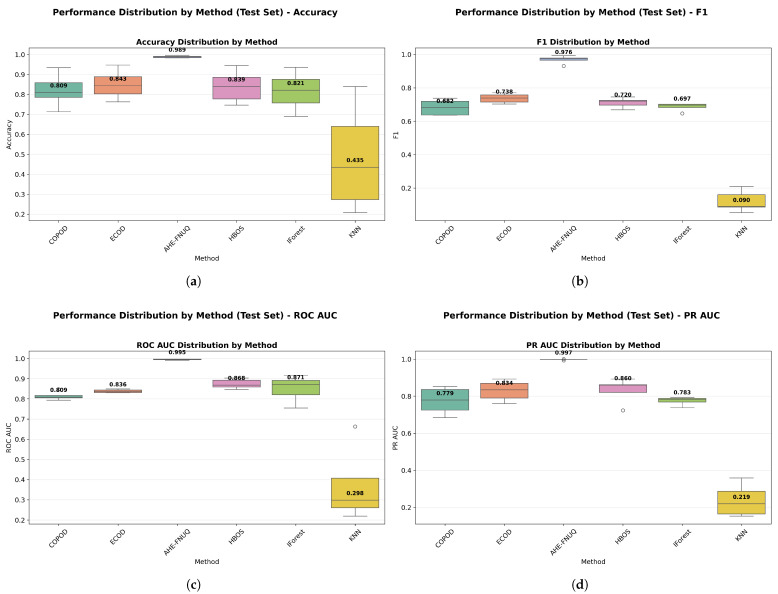
IoT agriculture 2024 dataset: Performance distribution by method—test set.

**Figure 35 sensors-25-06841-f035:**
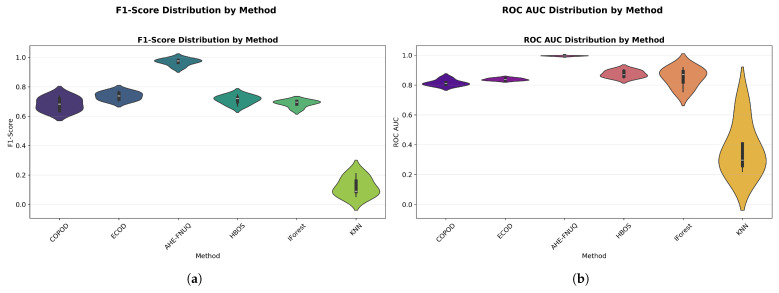
IoT Agriculture dataset: (**a**) ROC AUC distribution by method. (**b**) F1-score distribution by method.

**Figure 36 sensors-25-06841-f036:**
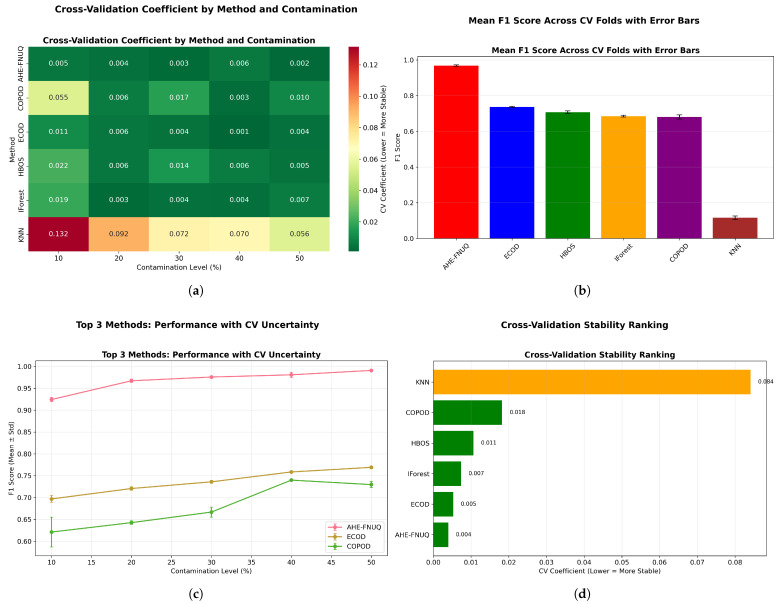
Cross-validation performance consistency analysis.

**Table 1 sensors-25-06841-t001:** Comparison of anomaly detection approaches across IoT domains.

Study	Domain	Method Type	Key Strengths	Limitations	Dataset Size
[40]	General IoT	Time-series Survey	IoT-specific focus	Use-case specificity	Medium
[41]	General IoT	ML/DL Survey	Comprehensive analysis	High dimensionality issues	Large-scale
[39]	Distributed IoT	Active Learning	Reduces labeled data dependency	Limited adaptability	Medium
[42]	IoT Networks	ML/DL Techniques	Network-focused analysis	Algorithm inaccuracy	Variable
[43]	Industrial IoT	Systematic Mapping	Sector-specific focus	Limited ML integration	84 studies
[44]	Industrial IoT	Classification Review	Industrial context	Data quality issues	Medium
[45]	Agricultural IoT	Implementation Study	Practical challenges	No specific algorithms	Field studies
[46]	Agricultural IoT	DL	Domain-specific analysis	Visual data reliance	Small

**Table 2 sensors-25-06841-t002:** Domain-specific challenges and requirements.

Challenge Category	General IoT [39,40,41,42]	Industrial IoT [43,44]	Agricultural IoT [45,47,48,49,50]
**Data Characteristics**			
Data Sparsity	Moderate	Low	High
Seasonal Variations	Low	None	Critical
Multi-modal Integration	Moderate	Low	High

**Environmental Factors **			
Controlled Environment	Variable	High	None
Weather Dependencies	Low	None	Critical
Biological Complexity	None	None	High

**Operational Constraints**			
Power Limitations	Moderate	Low	High
Network Connectivity	Variable	High	Limited
Cost Sensitivity	Moderate	Low	Critical

**Performance Requirements **			
Real-time Processing	High	Critical	Moderate
Accuracy Requirements	High	Critical	High
Interpretability	Moderate	High	Critical

**Table 3 sensors-25-06841-t003:** Research gaps and solution matrix.

Research Gap	Identified By	Impact Level	Current Solutions
Sparse Data Handling	[45]	Critical	Limited
Seasonal Adaptation	[46]	High	Manual adjustment
Uncertainty Quantification	[41]	High	Basic confidence
Cost Effectiveness	[45]	Critical	Expensive solutions

**Table 4 sensors-25-06841-t004:** Comparison of methodological approaches.

Method & Domain	Learning Mode	Adaptivity/ Drift Handling	Imbalance & Noise Handling	Explainability/ Uncertainty	Computational/ Deployment
ADSim [51], IDS	Online unsupervised	Similarity clustering; no drift triggers	None; assumes stable traffic	None	Moderate; needs stable networks
AEWAE [52], IoT	Online supervised	Global drift adaptation via PSO	Partial class weighting; noise-sensitive	None	Moderate–high; IoT cloud/fog
FS Ensemble [53], IDS	Offline supervised	None (static FS)	Indirect noise mitigation; no adaptation	None	Low; static offline
CAD [54], Cloud	Offline + deep	None	Sensitive to noise; no imbalance mechanism	None	Very high; GPU/cloud required
SKM-XGB [55], IDS	Offline supervised	None	SMOTE–KMeans balancing; labeled data	SHAP (static)	Moderate; batch offline
SDN-Stacking [56], 5G	Offline + deep	None	Limited imbalance handling	None	High; SDN/cloud only
Jeffrey [57], CPS	Offline hybrid	None	Domain heuristics; limited generalization	Partial	Low–moderate; manual tuning
**AHE–FNUQ (ours) **	**Offline training + streaming **	**Dual-threshold dynamic activation **	**RobustScaler; synthetic outliers **	**Integrated uncertainty; feature importance **	**Moderate; edge-optimized; no GPU **

## Data Availability

The original data presented in the study are openly available at https://github.com/ahmedamamou/ahe-fnuq-datasets (access on 23 September 2025).

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
