# Peer review of "AHE-FNUQ: An Advanced Hierarchical Ensemble Framework with Neural Network Fusion and Uncertainty Quantification for Outlier Detection in Agri-IoT"

_sensors, 2025, doi:10.3390/s25226841_

Round 1

Reviewer 1 Report

Comments and Suggestions for Authors

Dear Authors,

After reviewing the paper, I have some critical comments:
- The article is 39 pages long and contains a large number of graphs, tables, and repetitions. Most of the content consists of detailed descriptions of well-known algorithms (e.g., Isolation Forest, KNN, HBOS) or rewritten formulas from the literature. There is no real theoretical contribution.
- The authors generally declare that they propose a "framework for anomaly detection in Agri-IoT." Still, they do not precisely define the problem they are solving or why existing methods fail. There are no clearly formulated research questions or hypotheses, making the entire experiment descriptive rather than scientifically analytical.
- The authors suggest that their method "combines a neural network with ensemble and uncertainty quantification." Still, similar approaches have existed in the literature for years (e.g., Deep Ensemble Methods, Bayesian Neural Networks). They do not demonstrate how AHE-FNUQ differs fundamentally from existing methods or why it is "advanced" or "hierarchical." - No raw data or source code were presented, making it impossible to verify the results.
- The process of generating "artificial anomalies" (Algorithm 1) is arbitrary and unsupported by sources. It is unknown whether the generated data reflects actual anomalies in agriculture.
- The thresholds used (e.g., ROC AUC > 0.75) are chosen ad hoc, without theoretical justification.
- It is unknown how large the test dataset was after contamination, or how the absence of data leakage was ensured.
- All presented results are nearly perfect (AUC 0.99, F1 = 0.9, etc.) for three different datasets, which is unrealistic for real-world Agri-IoT tasks.
- Information on errors, inter-fold variance, and external validation is lacking. The authors limited themselves to a single Friedman test, which, given such high results, does not prove reliability, but rather uniformity of results.

FusionNN's mechanism is unclear. The architectural description (16-8-1) is trivial and doesn't justify incorporating a neural network into a system based on simple classifiers. There's no evidence that FusionNN improves performance over simple fusion rules (e.g., majority voting).

Overly firm conclusions from limited data. Three small datasets (Kaggle, GitHub) don't demonstrate generalisation, only the ability to fit simple data.

Sincerely

Author Response

We chose to upload the PDF file. 

Reviewer 2 Report

Comments and Suggestions for Authors

This paper proposes the AFE-FNUQ framework for anomaly detection in agricultural IoT systems. However, several issues need to be addressed:

  1. It is recommended to clearly elaborate on the real technical innovations of this method compared with existing or similar works, in order to highlight the contributions and novelty of the paper.
  2. Many detailed network parameters are not described; it is suggested to add relevant parameter information.
  3. The training and testing sets for FusionNN are not clearly separated. Are there any overlaps between training and testing samples? Is there a potential data leakage problem?
  4. Using artificially generated anomalies to construct evaluation data may result in poor generalization to real scenarios. The authors should explain why this approach does not cause overfitting or lack representativeness.
  5. It is recommended that the authors carefully check the references and standardize the citation format according to the journal’s requirements.
  6. Some abbreviations should be written out in full when first mentioned.
  7. Figure 11 has incorrect left/right labeling.
Comments on the Quality of English Language

The English could be improved to more clearly express the research.

Author Response

We chose to upload the PDF file. 

Reviewer 3 Report

Comments and Suggestions for Authors

The paper presents an interesting application of AI for anomaly detection in Agricultural Internet of Things (Agri-IoT). The contributions and novelty of the paper can be recognized. The research method is appropriate; the experimental results are extensively reported with sufficient details and discussions.  After reading the paper, I have the following comments:

1) Consider revising the paper title to include the aspect of ‘Agricultural Internet of Things (Agri-IoT)’.

2)  In the figure describing the general flowchart and the related discussion, please state clearly which sensor types and data formats are fused; using some icons or distinctive color codes to display their parts in the fusion is recommended to ease the reader’s perception.

3) regarding uncertainty quantification, I suggest adding an illustrative example in the form of graphical presentation to clearly demonstrate how uncertainty is quantified and propagated in the FusionNN.

4) Please clearly display the internal structure of the FusionNN because it is a critical component in the proposed framework. Clearly define the layers which fuse the input data and which part of the model is in charge of the meta-learning functions.

5) It is better to provide an illustrative example of how flagged anomaly and uncertainty estimates trigger responses in the system.

6) Consider adding more descriptions or discussions regarding ranked anomalies, feature importance visualization, and uncertainty map. These can help clarify the framework’s capability in practical scenarios.

7) Regarding the Algorithm 1, real-world anomalies may follow different statistical distributions, such as sharp changes. So, only scaling the data by standard deviation might not be ideal. How to deal with such issues?

8) Moreover, the algorithm works by individual sample, doesn’t it? How about operating with data sequence?, such as time-series IoT data.

9) Consider using more datasets for benchmarking:

Agriculture-Vision Dataset, Challenge and Workshop (CVPR 2020); https://github.com/SHI-Labs/Agriculture-Vision

AgroMind; https://github.com/rssysu/AgroMind

Smart Farming Sensor Data for Yield Prediction; https://www.kaggle.com/datasets/atharvasoundankar/smart-farming-sensor-data-for-yield-prediction

10) Figure 18. Green House: Performance Evaluation Splits by Method. Consider placing the metric value on top of the bar for better visualization of the results.

Author Response

We chose to upload the PDF file. 

Reviewer 4 Report

Comments and Suggestions for Authors

In this paper, the authors propose a framework to address the artificial agriculture problem, which includes data processing, anomaly detection, and prediction. Specifically, they adopt an ensemble-based approach for anomaly detection. To evaluate the effectiveness of their method, they conduct experiments on real-world agricultural datasets. The topic is quite innovative. After reading this paper, I have the following suggestions:

1st  In Section 3.3.2, the authors state that they adopt an ensemble layer for anomaly detection. However, they do not explain how the classifiers are combined in their method. I suggest that the authors clarify this part in more detail.

2nd  In 3.3.1, they introduce the way they preprocess the data. I suggest them to show a plot with real world example so people can get it easily.

3rd  In experiment, I suggest them to add a case study to show the effectiveness of their algorithm.

4th I suggest that the authors consider the presence of noise in the agricultural dataset and illustrate how it differs from other types of data.

5th Pay attention to grammar and spelling errors. Also, the format of figure 1 should be adjusted.

6 I suggest that the authors enrich the related work section on ensemble learning. I list several relevant papers below:

1 Chen, Ziheng, Jin Huang, Jiali Cheng, Yuchan Guo, Mengjie Wang, Lalitesh Morishetti, Kaushiki Nag, and Hadi Amiri. "FUTURE: Flexible Unlearning for Tree Ensemble." arXiv preprint arXiv:2508.21181 (2025).

2 Zaidi, Syed Ali Jafar, Attia Ghafoor, Jun Kim, Zeeshan Abbas, and Seung Won Lee. "HeartEnsembleNet: An innovative hybrid ensemble learning approach for cardiovascular risk prediction." In Healthcare, vol. 13, no. 5, p. 507. 2025.

Author Response

We chose to upload the PDF file. 

Round 2

Reviewer 1 Report

Comments and Suggestions for Authors

Dear Authors,

Thank you for your responses. After reviewing them, I can say that they have dispelled my initial doubts.

Sincerely.

Reviewer 3 Report

Comments and Suggestions for Authors

I have no further comments.